# Differentiation shifts from a reversible to an irreversible heterochromatin state at the DM1 locus

Tayma Handal[1,2,10], Sarah Juster[1,2,10], Manar Abu Diab[1,2,10], Shira Yanovsky-Dagan[1,2], Fouad Zahdeh[3], Uria Aviel[1,2], Roni Sarel-Gallily[4], Shir Michael[1,2], Ester Bnaya[1,2], Shulamit Sebban[5], Yosef Buganim[5], Yotam Drier [6], Vincent Mouly[7], Stefan Kubicek [8], Walther J. A. A. van den Broek[9], Derick G. Wansink [9] ✉, Silvina Epsztejn-Litman[1] & Rachel Eiges [1,2] ✉

Epigenetic defects caused by hereditary or de novo mutations are implicated in various human diseases. It remains uncertain whether correcting the underlying mutation can reverse these defects in patient cells. Here we show by the analysis of myotonic dystrophy type 1 (DM1)-related locus that in mutant human embryonic stem cells (hESCs), DNA methylation and H3K9me3 enrichments are completely abolished by repeat excision (CTG2000 expansion), whereas in patient myoblasts (CTG2600 expansion), repeat deletion fails to do so. This distinction between undifferentiated and differentiated cells arises during cell differentiation, and can be reversed by reprogramming of gene-edited myoblasts. We demonstrate that abnormal methylation in DM1 is distinctively maintained in the undifferentiated state by the activity of the de novo DNMTs (DNMT3b in tandem with DNMT3a). Overall, the findings highlight a crucial difference in heterochromatin maintenance between undifferentiated (sequence-dependent) and differentiated (sequence-independent) cells, thus underscoring the role of differentiation as a locking mechanism for repressive epigenetic modifications at the DM1 locus.

Epigenetic marks play a critical role in regulating chromatin structure and gene expression. The best documented and intensively studied epigenetic mark is DNA methylation. DNA methylation is associated with transcriptional silencing and plays an important role in genomic imprinting, X-chromosome inactivation, genomic instability, embryonic development and cancer[1]. The patterns of DNA methylation are generally set during embryo implantation and vary across different cell types and tissues[2]. They are determined by the coordinated and

[1]Stem Cell Research Laboratory, Medical Genetics Institute, The Eisenberg R&D Authority, Shaare Zedek Medical Center, Jerusalem 91031, Israel. [2]The Hebrew University School of Medicine, Jerusalem 91120, Israel. [3]Medical Genetics Institute, Shaare Zedek Medical Center, Jerusalem 91031, Israel. [4]The Azrieli Center for Stem Cells and Genetic Research, Department of Genetics, The Life Sciences Institute, The Hebrew University, Jerusalem 91904, Israel. [5]Department of Developmental Biology and Cancer Research, The Institute for Medical Research Israel-Canada, The Hebrew University-Hadassah Medical School, Jerusalem 91120, Israel. [6]The Lautenberg Center for Immunology and Cancer Research, IMRIC, Faculty of Medicine, The Hebrew University, Jerusalem, Israel. [7]Sorbonne Université, Inserm, Institut de Myologie, Centre de Recherche en Myologie, F-75013 Paris, France. [8]CeMM Research Center for Molecular Medicine of the Austrian Academy of Sciences, Lazarettgasse 14, AKH BT 25.3, 1090 Vienna, Austria. [9]Department of Medical BioSciences, Research Institute for Medical Innovation, Radboud University Medical Center, Nijmegen, The Netherlands. [10]These authors contributed equally: Tayma Handal, Sarah Juster, Manar Abu Diab. ✉e-mail: rick.wansink@radboudumc.nl; rachela@szmc.org.il

opposing activities of the methylating (DNMTs) and de-methylating (TETs) enzymes, with the help of additional factors in the cell, and can be passively lost during DNA replication[3]. However, little is known as to how methylation patterns are initially set during early embryonic development, or what guides the de novo DNMTs to specific regions in the genome. In addition, it is not fully clear how methylation patterns become fixed in the cell in a lineage-specific fashion, and how they are stringently maintained during the lifetime of the individual.

Rarely, errors in DNA methylation patterns occur. These types of mistakes are referred to as epimutations and are often secondary to germline transmitted or de novo mutations[3]. They provide a genetic basis for a variety of human developmental and neurological disorders (reviewed by[4]). Such defects may impede the setting of new methylation designs (i.e., de novo activity) or interfere with the copy of already established methylation patterns (i.e., maintenance activity). Whether secondary epimutations, including abnormal DNA methylation, can be reversed by the correction of the causative mutation in cells of patients has yet to be investigated. Although attempts thus far have been conducted to de-methylate the *FMR1* gene in cells with the fragile X syndrome repeat expansion via gene editing, these studies have been limited to undifferentiated iPSCs and produced somewhat mixed results[5,6].

The aim of this study was to explore whether the deletion of the underlying mutation for myotonic dystrophy type 1 (DM1) would restore the proper epigenetic status of the mutant locus. DM1 (OMIM 160900) is an autosomal dominant muscular dystrophy that results from a trinucleotide CTG repeat expansion (50 – >5000 triplets) in the 3′-UTR of the Dystrophia Myotonica Protein Kinase gene (*DMPK*)[7,8]. Although DM1 is primarily mediated by toxic RNA gain-of-function mechanisms[9], it is also characterized by *DMPK* hypermethylation and a gain of repressive histone post-translation modifications such as H3K9me3 and H3K27me3[10–15]. When the CTG number exceeds ~300, early in the course of development, the region surrounding the repeat becomes incorrectly de novo methylated in a way that depends on mutation size[14]. Specifically, the larger the expansion, the farther the aberrant methylation spreads from the upstream flanking region towards the repeat[14]. Although the significance of this methylation spread to disease pathology remains a topic of debate, *DMPK* hypermethylation provides the strongest indicator for the almost exclusive maternal transmission of the congenital and most severe form of the disease (CDM)[15]. Furthermore, it was suggested to be correlated with muscle strength and respiratory-related phenotypes[16], and may potentially explain the *cis* reduction in the transcription levels of the downstream close neighboring gene *SIX5*[17–23]. How precisely these lengthy CTG expansions lead to de novo methylation of the 3′-end of *DMPK* remains unknown. One potential mechanism is the loss of CTCF binding immediately upstream to the CTGs[24]. However, data from transgenic mice and mutant hESCs argue against this supposition[13,14]. In addition, it is still unclear whether hypermethylation, once established, can be reversed in affected cells.

Here, we examined whether complete excision of the CTG expansion from the *DMPK* gene (>1000 repeat units) would abolish DNA hypermethylation and heterochromatinization and if so, whether it relies on the differentiation state of the cell. We show that the excision of CTGs in undifferentiated hESCs (CTG2000) resets the locus by abolishing DNA methylation and H3K9me3 enrichment. This is in sharp contrast to the effect of repeat deletion in affected myoblasts (CTG2600), where the repressive epigenetic modifications remained unchanged. Furthermore, we provide evidence for a switch from reversible to fixed aberrant methylation by in vivo differentiation of hESCs, which can be set back in iPSCs by the reprogramming of DM1-affected gene-edited myoblasts. Taken together, these findings in DM1 suggest that the repair of the mutation in cells of patients may not be sufficient to therapeutically address the epigenetic aspects of the disease. This may apply to a wide range of genetically transmitted disorders that coincide with aberrant epigenetic modifications.

## Results

### Deletion of a large CTG repeat expansion from *DMPK* in DM1 hESCs

To examine whether *DMPK* hypermethylation could be reversed in DM1-affected human embryonic stem cells (hESCs), we excised a CTG2000 expansion from a mutant hESC line (SZ-DM14, 5/2000 CTGs) with a heavily methylated allele (-100%)[14]. Using a pair of guide RNAs (gRNAs) designed to target positions −10 bp and +47 bp relative to the repeat sequence (Fig. S1a), we achieved complete removal of the CTG, minimizing alteration in the flanking sequences without altering putative binding sites for CTCF. Validation of repeat deletion from one or both alleles was conducted by Gene Scan analysis (Fig. S1b) followed by a PCR assay overlapping the 5′ breakpoint next to the CTG (Fig. S1c). DNA Sanger sequencing was then utilized to confirm precise repair of the double-strand DNA breaks following repeat excision. Subsequently, selected clones demonstrating successful repeat elimination underwent further validation through Southern blot analysis following enzyme restriction, leaving no room for uncertainty regarding the effective editing of the wild-type and/or mutant allele within the cell (refer to Fig. S2a).

Of the 18 targeted clones established (see Table S1), two were completely *DMPK* repeat deficient (Δ/Δ). While in one clone the sequence was perfectly repaired with no indels in the normal and expanded alleles (CL9), in the other clone (CL29), one allele was perfectly repaired while the other had a 2 bp deletion at the junction, thus providing compelling evidence that CL9 and CL29 originated from distinctly different clones (Fig. S1d). In the remaining clones, targeting was less efficient and led to a more complex and unpredictable result (Table S1), consistent with our previous findings[25]. To address concerns about potential off-target effects elsewhere in the genome, we Sanger sequenced candidate sites according to the WTSI Genome Editing tool. The results suggest that no cleavage activity outside of the *DMPK* locus occurred in both Δ/Δ clones, as corroborated by this assay (Fig. S2b).

### Repeat excision reverses aberrant *DMPK* methylation and heterochromatinization in mutant hESCs

We examined whether the elimination of the expanded repeat had reversed aberrant methylation patterns upstream to the CTGs. To do this, we measured DNA methylation levels before and after repeat excision using bisulfite DNA colony sequencing. Methylation levels were measured nine passages after gene manipulation by colony bisulfite sequencing 650 bp upstream from the CTG repeat (26 CpG sites). This region was previously identified to be part of a disease-associated DMR which is hypermethylated in hESCs when the repeat expands beyond 300 triplets in a way that depends on expansion size[14]. This experimental approach provided clear evidence for a widespread event of demethylation from 55% (unmanipulated parental cells, corresponding to 100% methylation on the mutant allele) to 0% in both of the Δ/Δ CTG-deficient clones (CL9 and CL29). Additionally, it demonstrated 0% methylation in the unaffected hESC line control (SZ-RB26), both before and after gene editing (Fig. 1a). Locus-specific bisulfite deep-sequencing in an overlapping region (15 CpGs, 759 bp to 631 bp upstream relative to the repeat) in the parental cell line and Δ/Δ clones was used to unequivocally show that methylation was not preserved in any of the successfully edited molecules (Fig. S3a). This contrasted with gene-edited hESC clones CL7 and CL13, in which repeat contraction either resulted in no change, or reduced aberrant methylation levels, respectively (Fig. 1a). This methylation analysis at the DMR in the repeat-targeted clones suggests that *DMPK* hypermethylation in DM1 depends on the constant presence of the expansion mutation in undifferentiated hESCs.

Next, to explore whether hypomethylation is coupled with the loss of heterochromatin, we analyzed the enrichment of two repressive

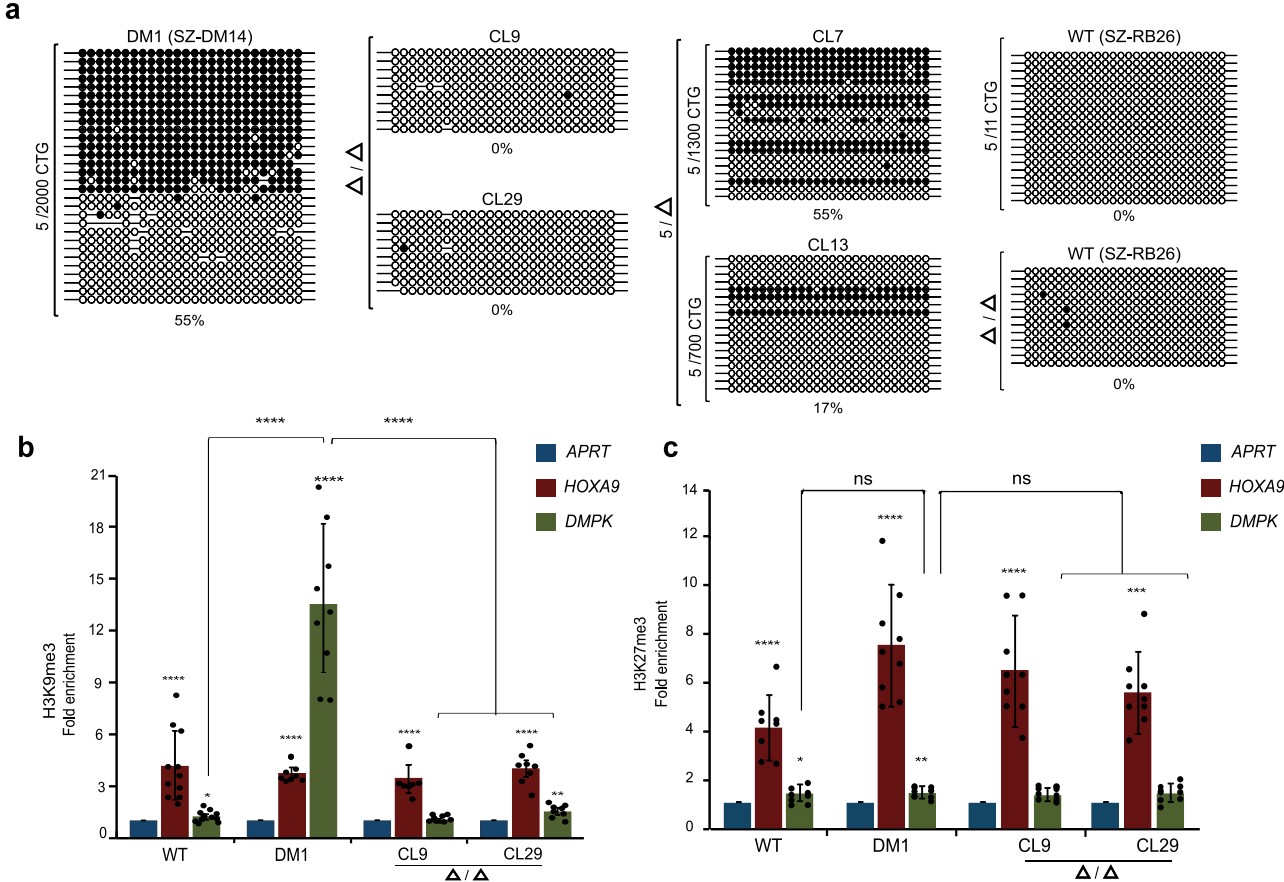

**Fig. 1 | Reversal of abnormal methylation and loss of repressive histone modifications by CTG repeat excision in mutant hESCs. a** Colony DNA bisulfite sequencing of the DMR (488-777 bp upstream of the repeat, 26 CpG sites) in unmanipulated DM1 hESCs with a heavily methylated CTG2000 expansion (SZ-DM14, methylation levels of 55%), a pair of gene-edited CTG-deficient (Δ/Δ) clones (CL9 and CL29, methylation levels of 0%) and two CRISPRed clones still bearing 1300 CTGs (55%, CL7) and 700 and fewer CTGs (17%, CL13), and wild type hESCs with 5/11 CTG alleles (SZ-RB26, methylation levels of 0%) before and after (Δ/Δ) repeat excision. Filled circles: methylated CpGs; empty circles: unmethylated CpGs. **b** Real-time PCR ChIP analysis for H3K9me3 in wild type (SZ-13), parental DM1 affected hESC line (SZ-DM14, CTG2000), and isogenic CTG-deficient homozygote clones (CL9 and CL29). *APRT* and *HOXA9* were used as negative and positive controls, respectively. Negative controls were set to one. The data is derived from either *n* = 3 (SZ-DM14, CL9, and CL29) or *n* = 4 (wild type) independent ChIP experiments. Each panel illustrates the average ± standard deviation (STD)

calculated across all technical replicates. Statistically significant enrichments were calculated within each cell line for *DMPK* and *HOXA9* by comparing to *APRT*, and between cell lines for *DMPK* by pairwise comparison to DM1-affected hESC line (two-sided paired *t*-test). **c** Real-time PCR ChIP analysis for H3K27me3 in wild type (SZ-13), parental DM1 affected hESC line (SZ-DM14, CTG2000), and isogenic CTG-deficient homozygote clones (CL9 and CL29). *APRT* and *HOXA9* were used as negative and positive controls, respectively. Negative controls were set to one. The data is derived from *n* = 3 independent ChIP experiments. Each panel illustrates the average ± standard deviation (STD) calculated across all technical replicates. Statistically significant enrichments were calculated within each cell line for *DMPK* and *HOXA9* by comparing to *APRT* by pairwise comparison to DM1-affected hESC line (two-sided paired *t*-test). *P*-values: ns = *p* > 0.05 **p* < 0.05, ***p* < 0.01, ****p* < 0.001, *****p* < 0.0001. Precise *P*-values are provided in Table S4. Source data are provided as a Source data file.

histone modifications: H3K9me3 (representing constitutive heterochromatin) and H3K27me3 (representing facultative heterochromatin) immediately upstream to the DM1 repeat, by chromatin immunoprecipitation (ChIP). While H3K9me3 was exclusively enriched in the unmanipulated DM1 hESCs (Fig. 1b), neither of the hESCs lines/clones were enriched for H3K27me3 (Fig. 1c). Hence, H3K9me3 deposition, but not H3K27me3, is tightly correlated with CTG array size and the gain of aberrant methylation in hESCs. These results indicate that repeat removal in mutant hESCs alters the epigenetic status of the locus in a way that prevents constitutive heterochromatin from being re-established near the repeat.

**Altered chromatin structure likely restores CTCF binding but does not affect local gene transcription**

It has been suggested that aberrant methylation patterns upstream of the CTGs might alter the chromatin structure through the loss of CTCF binding. Consistent with this claim, we determined methylation levels

by bisulfite colony sequencing immediately upstream to the repeat (region F in ref. 14) and validated the binding of CTCF in an overlapping region (CTCF binding site I, CTCFI) by ChIP analysis in wild type, DM1 and Δ/Δ hESCs (Fig. S3b and S3c). Although we were unable to distinguish between the wild type and affected hESCs in terms of extent of enrichment using the ChIP assay (maximum 2-fold change), we found that hypermethylation at the CTCF binding site was exclusive to DM1 hESCs (Fig. S3b). In addition, we leveraged the 2 bp deletion that was induced by gene editing at the junction of one of the alleles in CL29 (unmethylated Δ/Δ clone) to show the presence of two alleles in the CTCF bound fraction after ChIP experiment (Fig. S3d). This, together with the known role of methylation in abolishing CTCF binding next to the repeat in hESCs (Fig. S5D and S5E in ref. 14), and the clear evidence that gene editing does not disrupt CTCF binding next to the CTG (as illustrated by substantial enrichments in Δ/Δ clones, Fig. S3c), strongly suggests that the loss of heterochromatin by repeat deletion restores CTCF occupancy in the mutant allele.

It has been claimed that the change in chromatin structure may alter local gene transcription at the DM1 locus[14,26,27]. Therefore, we assessed the total mRNA levels of *DMPK* and *SIX5* in the wild type, DM1-affected and DM1-Δ/Δ hESCs by RT-ddPCR. The results showed that the wild type and affected DM1 hESCs did not significantly differ for total *DMPK* and *SIX5* mRNA levels (Fig. S3e). Nor could we find a significant change in *SIX5* mRNA levels or a consistent trend in the expression of *DMPK*, when comparing DM1 unmanipulated vs. gene-edited hESCs (Fig. S3e). Thus, in conjunction with our published data reporting no change in *DMPK* and *SIX5* mRNA expression levels in DM1 unmanipulated as compared to gene-edited myoblasts (see ref. 25), this strongly suggests that local gene expression is unaffected by abnormal methylation at the DM1 locus, at least not in embryonic stem cells and patient myoblasts.

## CTG repeat excision does not restore the normal epigenetic status of the locus in patient myoblasts

We explored the relevance of our findings on *DMPK* demethylation by gene editing to patients' cells. To do this, we utilized previously established patient-derived repeat-deficient myoblasts[25]. Since the CTG2600 repeat in these cells was targeted with nearly the same pair of gRNAs as for the hESCs (see Fig. S1a, marked by green asterisks), these cells provided a good opportunity to compare the effect of expanded repeat excision on the methylation status of the DM1 locus in myoblasts vs. undifferentiated hESCs. The analysis of DNA methylation levels in myoblast clones with and without the CTG2600 repeat was carried out precisely as described for the hESCs (i.e. colony bisulfite sequencing 650 bp away from the repeat, 26 CpG sites) after at least 20 population cell doublings. In this case, however, we utilized a non-CpG informative SNP within the DMR to perform allele-specific methylation analysis (rs635299, see also ref. 14). This allowed us to easily distinguish between normal (CTG13; variant G) and expanded (CTG2600; variant T) alleles during the methylation analysis. This approach demonstrated that methylation levels remained unchanged after the complete deletion of the CTG repeat from *DMPK* in three homozygous (Δ/Δ) and one heterozygous (13/Δ) myoblast clones and presented levels of 100% on the background of the mutant allele (T variant) (Figs. 2a and S4). This was found in addition to the absence of change in methylation levels in three other independent clones, where gene editing was inefficient and failed to remove the CTG repeat from either allele (CTG13/CTG2600). In no case were normal alleles hypermethylated in the wild type control myoblasts or in the mutant myoblasts against the background of the normal allele (variant G, based on the analysis of 150 wild type molecules in total, data not shown), thus ruling out the possibility of non-physiological hypermethylation due to culture conditions.

To explore whether hypermethylation was coordinated with heterochromatin, we confirmed significant enrichment for H3K9me3 by ChIP analysis in all of the affected (successfully and unsuccessfully gene-edited) myoblast clones (Fig. 2b). Strikingly however, when we monitored for H3K27me3, we observed significant enrichment levels in all the examined cell clones, before and after editing. This suggests that H3K27me3 is elicited with heterochromatinization in a way that depends on differentiation into myoblasts. Overall, the results of this analysis lead to the conclusion that the removal of the expansion in DM1-affected myoblasts cannot reset the normal epigenetic status of the DM1 locus once heterochromatin has been established.

Finally, to explore whether persistent methylation in the gene-edited myoblasts could be removed by treatment with a demethylating agent, we monitored for aberrant methylation levels in four different edited clones after a 3 day 5-Aza-dC treatment (5 μM). Strikingly, in three out of the four tested clones, the methylation levels remained unchanged at the DMR (100% against the background of the mutant allele, Fig. S4). In the one remaining clone, methylation levels were reduced from 88% to 55% (M4). None of the methylation levels

decreased in the parental DM1 myoblasts as a result of drug treatment (M1). Thus, in most cases (3/4 edited clones), 5-Aza-dC treatment could not restore the normal epigenetic status of the locus against the background of the mutant allele, despite the excision of the repeat.

## Differentiation elicits irreversible *DMPK* hypermethylation in DM1 hESCs

Given the marked differences between the DM1 hESCs and affected myoblasts in terms of the ability to epigenetically reset the locus after gene editing, we next investigated whether differentiation would indeed abolish the reversibility of hypermethylation. For this purpose, we induced the mutant hESCs (SZ-DM14, CTG2000) to spontaneously differentiate in vivo by producing teratomas (benign tumors that contain tissues representatives of all three germ layers) in immuno-compromised mice. These were then used to generate a more homogenous cell population by establishing fibroblast-like cell cultures termed TOFs (teratoma-derived fibroblasts). We chose to target the repeats specifically in TOFs, since the resulting cell cultures are highly homogeneous, can be grown to large numbers, and can easily be transfected, which makes gene editing much more efficient and easier to achieve. Using precisely the same gRNAs as described above for targeting the repeats in hESCs, we removed the CTGs from wild type and expanded alleles (nearly 100% efficiency, see Fig. S5) following cell differentiation. Similar to the results for the edited myoblasts, the excision of the repeats from TOFs did not change the levels of abnormal methylation (45% vs 42%, before and after gene editing, respectively) (Fig. 3a). This provides evidence for the inhibitory effect of differentiation on the reversibility of this process and excludes the possibility of a myogenesis-specific molecular event.

## Reprogramming to pluripotency facilitates reversible methylation in affected gene-edited myoblasts

To further substantiate the effect of differentiation on the reversibility of this process, we explored whether the conversion of hypermethylated CTG-deficient myoblasts into iPSCs would restore the normal epigenetic status of the locus. For this purpose, we reprogrammed a pair of myoblast clones that were successfully targeted and accurately repaired (Δ/Δ, clones M4-IPSC2/IPSC3 (derived from M4 edited myoblasts) and M6-IPSC4/IPSC5/IPSC6 (derived from M6 edited myoblasts)) (Fig. S6a). Analysis of DNA methylation levels in the resulting iPSCs was carried out as described for the gene-edited myoblasts by utilizing a non-CpG informative SNP within the DMR (rs635299). This procedure confirmed that reprogramming the gene-edited myoblasts into iPSCs reduced abnormal methylation, albeit to a variable levels in different clones (Fig. 3b and S6b), reaching practically 0% in one of them (M4-iPSC3). This contrasts sharply with the observations in iPSCs with an intact mutation (M1-IPSC1), where methylation levels never change (100% on the background of the mutation). These results demonstrate that the erasure of hypermethylation via the transition to a pluripotent state is conditioned by a preceding step of repeat removal.

Given the extensive cell proliferation of all the investigated cell types (i.e., hESCs, TOFs, myoblasts and iPSCs) it appears very unlikely that de-methylation in the gene-edited pluripotent cells resulted from passive dilution during DNA replication, as opposed to active removal by de-methylating enzymes. Thus, combined with the findings above, we provide evidence for a transition from a reversible to a fixed heterochromatin state at the DM1-expanded locus, which is elicited by differentiation and could be set back by somatic cell reprogramming.

## Abnormal methylation at the DM1 locus is maintained by de novo DNMT activity in hESCs

Given the abovementioned results, we aimed to identify chromatin modifiers that would explain the difference in the maintenance of

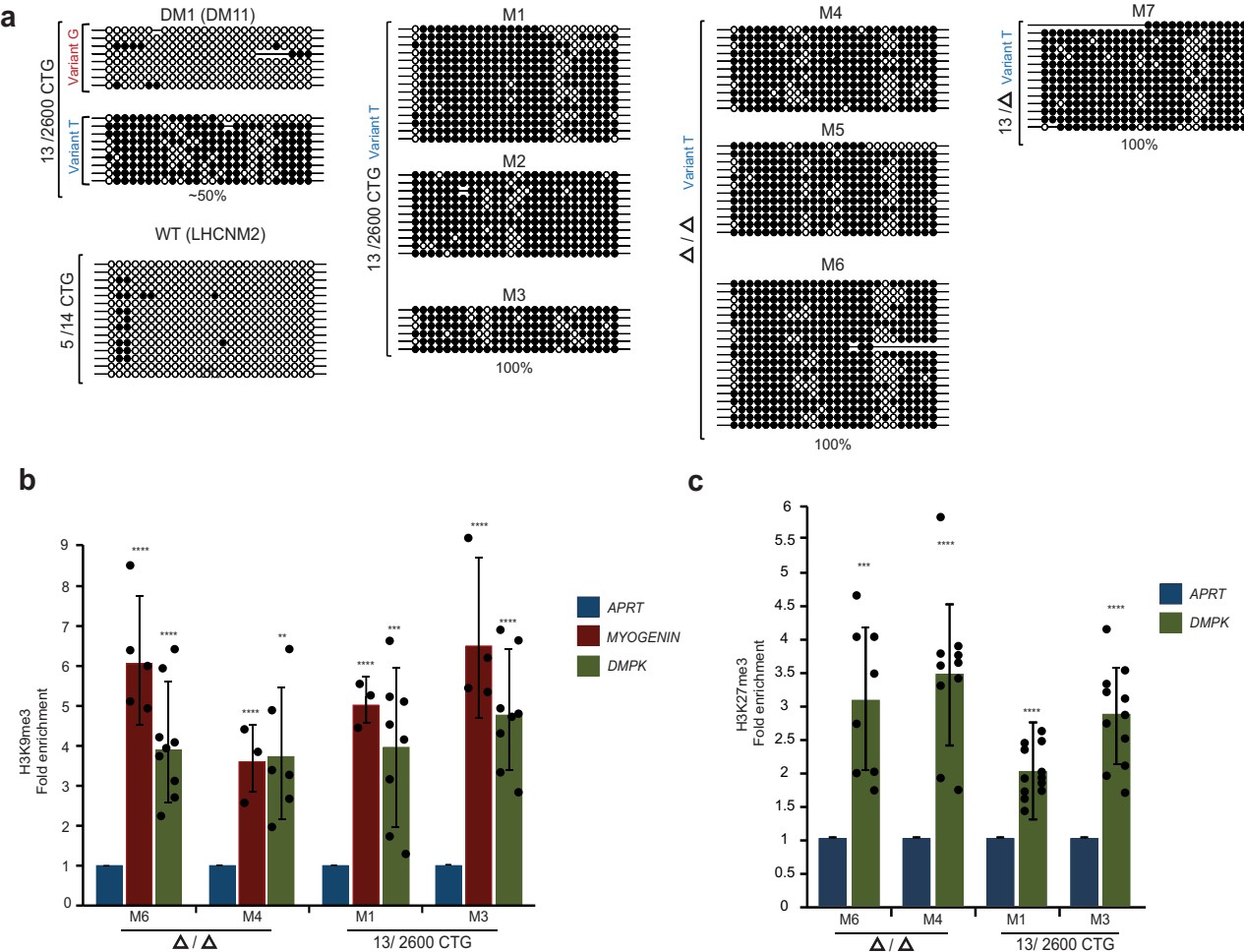

**Fig. 2 | CTG excision in affected myoblasts does not restore the normal epigenetic status of the DM1 locus. a** Allele-specific colony DNA bisulfite sequencing at the disease-related DMR (488-777 bp upstream of the repeat, 26 CpG sites) pre and post repeat excision (after >20 population cell doublings) from affected myoblasts with normal and expanded alleles (13/2600 CTG), three completely CTG-deficient (Δ/Δ), one heterozygote for the deletion against the mutant allele (13/Δ), three unsuccessfully manipulated clones (13/2600 CTG), and an independent control myoblast cell line (5/14 CTG). Methylation patterns shown for manipulated clones (Δ/Δ, 13/Δ, and 13/2600) on variant T background. Filled circles: methylated CpGs; empty circles: unmethylated CpGs. **b** ChIP analysis for H3K9me3 in successfully edited (Δ/Δ, clones M4 and M6) versus unsuccessfully edited (13/2600 CTG, clones M1 and M3) DM1 myoblast clones. *APRT* and *MYOGENIN* as controls. Data: derived from *n* = 3 (M4) or *n* = 4 (M6, M1 and M3) independent ChIP

experiments. Each panel illustrates the average ± standard deviation (STD) calculated across all technical replicates. Statistically significant enrichments calculated within each cell line for *DMPK* and *MYOGENIN* compared to *APRT* (two-sided paired *t*-test). **c** ChIP analysis for H3K27me3 in successfully edited (Δ/Δ, clones M4 and M6) versus unsuccessfully edited (13/2600 CTG, clones M1 and M3) DM1 myoblasts. *APRT* as negative control, set to one. Data: derived from *n* = 2 (M6), *n* = 3 (M4 and M3) or *n* = 5 (M1) independent ChIP experiments. Each panel illustrates the average ± standard deviation (STD) calculated across all technical replicates. Statistically significant enrichments calculated within each cell line for *DMPK* compared to *APRT* (two-sided paired *t*-test). *P*-values: *$p < 0.05$, **$p < 0.01$, ***$p < 0.001$, ****$p < 0.0001$. Precise *P*-values are provided in Table S4. Source data are provided as a Source data file.

abnormal methylation at the DM1 locus between undifferentiated and differentiated cells.

First, we profiled gene expression by RNA-seq and compared the undifferentiated DM1 hESCs to their differentiated counterparts, teratoma-derived fibroblasts (TOFs). Furthermore, we extended the analysis to include a comparison of undifferentiated hESCs to the patients' myoblasts, to further substantiate our findings.

After validating the exclusive expression levels of the pluripotent-specific markers *POU5F1* (*OCT4*), *NANOG* and *SOX2* in the hESCs, we compared the expression of nearly 150 potentially relevant genes, collectively representing the complete repertoire of chromatin modifiers in the genome (adapted from dbEM with few additions, see link at http://web.iiitd.edu.in/rghava/dbem/ and ref. 28) (Supplementary Data 1). To visualize the significant differences in the expression of potential chromatin modifiers between the two cell states (hESCs vs.

TOFs/myoblasts), Volcano plots were generated (Fig. 4a). Based on this analysis, the de novo DNA methyltransferases (*DNMT3a and DNMT3b*) and the de-methylating enzymes *TET1* and *TET2* emerged among the most significant. While *DNMT3b* and *TET1* exhibited exclusive expression levels in the undifferentiated hESCs, *DNMT3a* and *TET2* demonstrated a marked change, with either increased or reduced expression in the hESCs, respectively. This contrasted with the expression patterns of *DNMT1* and *TET3*, both of which maintained comparable mRNA levels across all cell types and states.

The notable similarity in the differential expression of these enzymes between the two Volcano plots encouraged us to conduct functional assays. First, we chose to knock out *DNMT3b*, because *DNMT3b* stood out as the most statistically significant, up-regulated and exclusively expressed chromatin modifier gene of all the hESCs on the list of differentially expressed genes (DEGs) identified (Fig. 4a). By

**a**

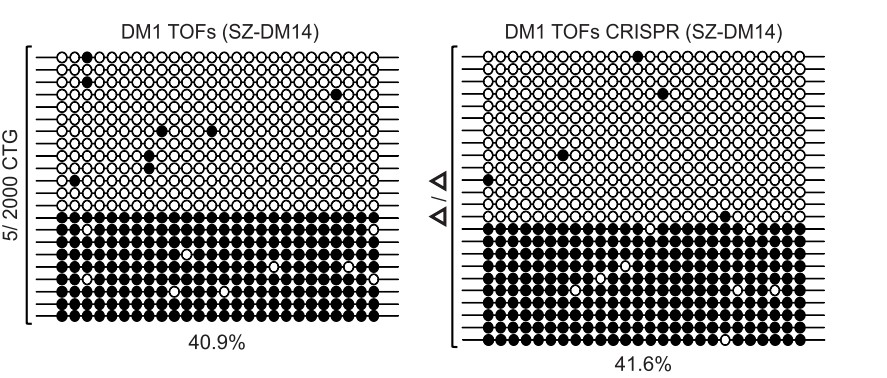

**b**

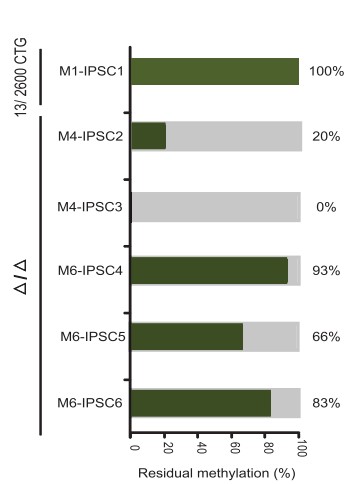

Fig. 3 | **The shift from a reversible to an irreversible heterochromatin state by hESC differentiation that can be set back by reprogramming after repeat removal. a** Colony DNA bisulfite sequencing of the DMR (26 CpG sites) in unmanipulated DM1 affected teratoma-derived cell cultures with a heavily methylated CTG2000 expansion (SZ-DM14 TOFs) before (DM1 Δ/Δ TOFs, 40.9%), and after (DM1 Δ/Δ TOFs CRISPR, 41.6%) repeat excision. Filled circles: methylated CpGs; empty circles: unmethylated CpGs. **b** A graph summarizing the aberrant methylation levels in iPSCs derived from successfully targeted (M4-IPSC2/3 and M6-IPSC4/5/6, Δ/Δ) and unsuccessfully edited (M1-IPSC1, 13/2600CTG) DM1 affected myoblast clones. Methylation levels at the DMR, against the background of the mutant allele (variant T) were determined by colony DNA bisulfite sequencing (also shown in Fig. S6b), ranging from 0% to 93% in the resulting Δ/Δ iPSCs.

inducing a pair of DSBs with two gRNAs using the CRISPR/Cas9 system, we introduced a homozygous 179 bp deletion overlapping the intron 1-exon 2 boundary in the *DNMT3b* gene (as depicted in Fig. 4b). We chose to specifically target exon 2, because it is shared by many different mRNA isoforms of the gene (at least 8 different isoforms for *DNMT3b*). Furthermore, by targeting the 5′-end of the coding region, we increased the probability of introducing premature termination codons (PTCs) or triggering mRNA degradation by nonsense-mediated decay (NMD).

After screening for bi-allelic deletions by PCR (Fig. S7a), the potential knockout clones (KOs) were Sanger sequenced (Fig. S7b) and then validated by Western blot analysis (Fig. 4b). These assays confirmed the complete elimination of the DNMT3b enzyme in 7 out of the 30 transiently selected hESC clones on the background of the DM1 hypermethylated allele.

To evaluate the effect of *DNMT3b* knockout (KO) on aberrant methylation in these cells, we measured the methylation levels at the disease-associated DMR in three randomly selected clones using bisulfite locus-specific DNA deep-sequencing (at least nine cell passages following gene manipulation). However, the *DNMT3b*-null clones did not exhibit any discriminable change in aberrant methylation levels when compared to the unmanipulated matched control (Fig. 4c). Given these results, we addressed the possibility of a functional overlap between DNMT3b and DNMT3a enzymatic activity in undifferentiated hESCs, which has been observed in multiple genomic regions[29]. For this purpose, we targeted *DNMT3a* on the background of pre-existing *DNMT3b*-null DM1 hESCs. Using the CRISPR/Cas9 editing system with a pair of gRNAs, we introduced a 128 bp deletion that overlapped with the boundary between exon 19 and the following intron in the *DNMT3a* gene (Fig. 4d). This approach was chosen for two main reasons. The first is that *DNMT3a* exhibits multiple isoforms (at least 6) arising from alternative transcription initiation and splicing sites, most of which involve the 3′-end of the gene (exons 7–23 region). The second is that exon 19 is critical for the catalytic activity of the enzyme (exons 16–20), in that it facilitates the comprehensive elimination of potentially active protein species[30]. Based on these rationales, we introduced a bi-allelic deletion in *DNMT3a* on the genetic background of DM1

*DNMT3b* KO hESCs in 5 out of 60 transiently selected puro-resistant clones (Fig. S7c).

After screening for homozygote deletions by PCR and Sanger sequencing (Fig. S7d, e), we performed quantitative reverse transcription PCR (RT-qPCR) to search for a substantial reduction in *DNMT3a* mRNA levels (Fig. 4d). We found that gene manipulation indeed led to either the complete loss (Cl-I3) or, more commonly, a significant reduction in *DNMT3a* mRNA levels (Fig. 4d). However, confirming the absence of DNMT3a enzymatic activity in these cells was difficult due to the absence of a straightforward assay for DNMT3a-specific activity. Nonetheless, given the pivotal role of exon 19 in the catalytic domain of DNMT3a[30], the residual transcripts were likely to be non-functional because of exon 19 skipping, as validated by Sanger sequencing of the RT-PCR products (Fig. 4d and Fig. S7e). In line with this view, we generated five different bi-allelic *DNMT3*- double targeted DM1 hESCs.

To assess the effect of de novo *DNMT3* double targeting, we monitored the levels of abnormal methylation within the disease-associated DMR by utilizing bisulfite locus-specific DNA deep-sequencing, as described above for single *DNMT3b* KO clones. There was a significant reduction in abnormal methylation levels in four out of the five assayed clones, ranging from 18% to 38.5% (out of a maximum of 50%, Fig. 4e). Crucially, note that unlike the single *DNMT3b* KOs, most of the double targeted clones became morphologically abnormal, tended to spontaneously differentiate, and proved to be incapable of sustaining growth beyond five passages.

In summary, this study revealed a fundamental difference between undifferentiated and differentiated cells in terms of the role played by de novo DNMTs (DNMT3a singly or jointly with DNMT3b) in maintaining abnormal methylation patterns at the DM1 locus in undifferentiated hESCs. Furthermore, and unlike in differentiated cells, this molecular event is dependent on the DNA sequence; i.e., the disease-causing CTG expansion at the *DMPK* gene.

## Discussion

We report complete removal of the CTG repeat from the *DMPK* gene in DM1 affected cells with particularly large expansions (>2000 repeat copies). By completely deleting the CTGs from DM1 undifferentiated

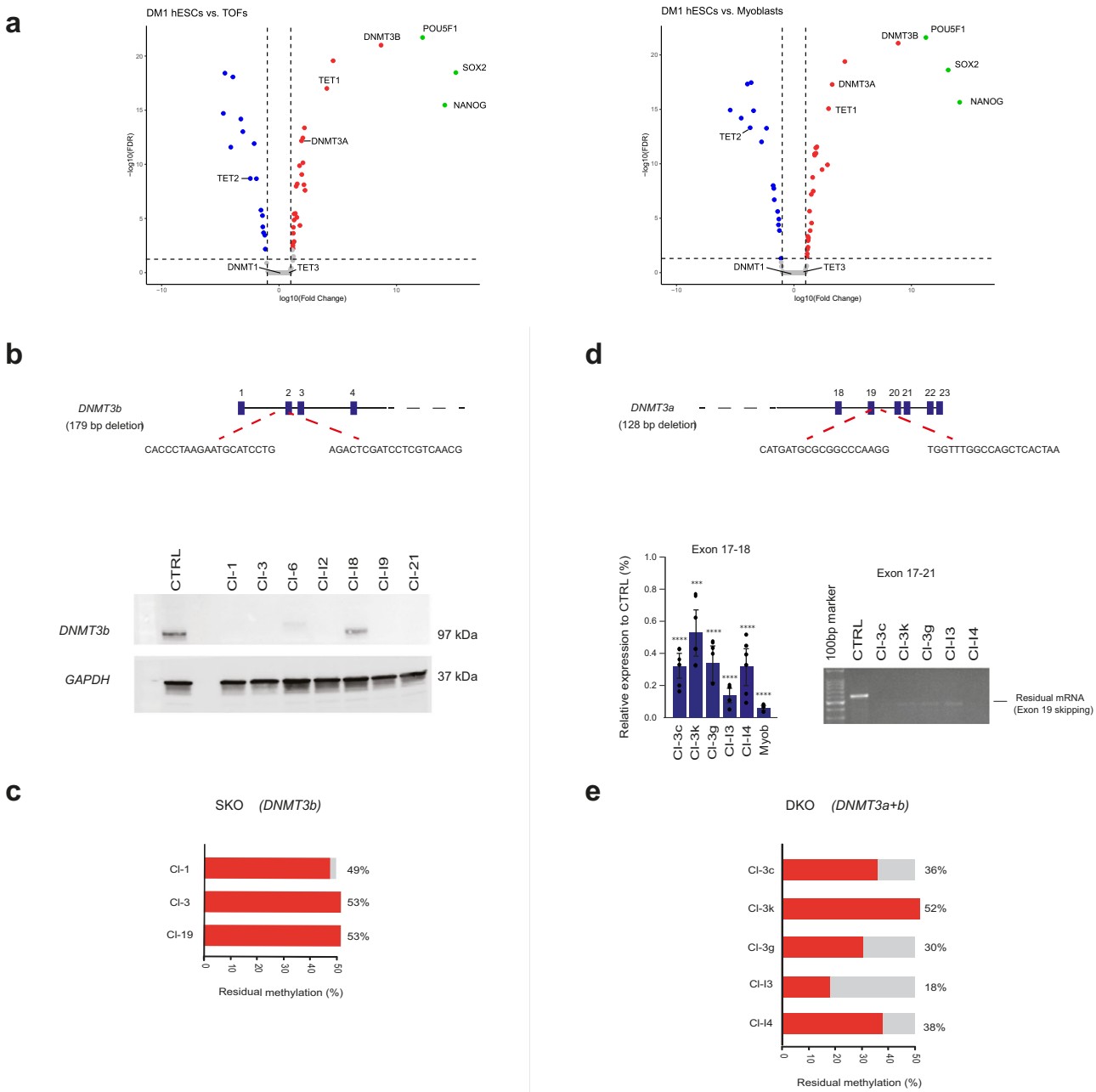

**Fig. 4 | Abnormal methylation at the DM1 locus is maintained by de novo DNMTs activity in hESCs. a** Volcano plots comparing chromatin modifier gene expression between undifferentiated DM1 hESCs and their in vivo differentiated counterparts: teratoma-derived fibroblasts (left) or patient myoblasts (right). Red denotes high expression in undifferentiated hESCs, blue in the alternative cell type (TOFs/myoblasts), and gray for equal expression. Green denotes the levels of three pluripotent-associated markers in undifferentiated hESCs. Each data set averages 3 technical experiments. Volcano plot analysis employed edgeR for RNA-seq analysis, with subsequent FDR correction for multiple testing. Plots were generated using the VolcanoNose R program. **b** *DNMT3b* targeting approach overview. Bottom: Western blot assesses DNMT3b protein levels in parental DM1-affected hESC line (CTRL) and genetically manipulated isogenic clones, with GAPDH as loading control. The experiment was conducted once. **c** Residual methylation levels (%) at the DM1-related DMR in single knockouts (SKO) of *DNMT3b* DM1 hESC clones determined via locus-specific bisulfite DNA deep-sequencing. Levels are relative to baseline in parental hESCs (SZ-DM14), set at 50%. **d** *DNMT3a* targeting approach overview. Bottom: *DNMT3a* mRNA levels assessed before and after gene editing in DNMT3-null DM1-affected hESCs. Residual *DNMT3a* transcripts post-editing examined by RT-qPCR (exon 17–18, left), and exon 19 skipping validation by RT-PCR (right). CTRL DM1-affected hESCs and M1 myoblasts (myob) served as positive and negative controls, respectively. Data per clone averages *n* = 4 (I3), *n* = 5 (3G) or *n* = 6 (I4, 3K, 3C) technical experiments. Error bars: standard deviation. Significant *DNMT3a* transcription changes assessed via pairwise comparison (two-sided paired *t*-test). *P*-values: ***$p < 0.001$, ****$p < 0.0001$. Precise *P*-values are provided in Table S4. **e** Residual methylation levels (%) in DM1-related DMR of double-targeted *DNMT3a* and *DNMT3b* DM1 hESC clones (DKO) determined via locus-specific bisulfite DNA deep-sequencing. Levels are relative to parental hESCs (SZ-DM14), set at 50%. Source data are provided as a Source data file.

and differentiated cells, we addressed the question of whether aberrant epigenetic modifications that are secondary to a disease-causing mutation can be erased by gene correction and if so, whether this is equally effective in different differentiation states.

We show that abnormal methylation and H3K9me3 enrichments are completely abolished by repeat excision in undifferentiated DM1-affected hESCs, providing evidence for a switch from a closed (heterochromatin) to an open (euchromatin) chromatin configuration

corresponding, at least in part, with similar experiments in iPSCs from fragile X syndrome[5,6]. This is most likely coupled with the rescue of CTCF binding near the repeat (Fig. S3c, d). Nevertheless, we could not find evidence for a significant change in *SIX5* mRNA levels or consistent trends in *DMPK* mRNA expression, before or after CTG excision (Fig. S3e). This was also true between wild type and DM1 affected hESCs and contrasts previous studies regarding the role for hypermethylation in regulating local gene transcription in DM1[14,26,27].

Strikingly though, when our experiment was replicated in DM1-affected myoblasts (CTG2600), repeat removal failed to restore the epigenetic status of the locus despite many population doublings in culture. This was exhibited, without exceptions, by the methylation levels of 100% in all the tested molecules on the background of the mutant allele (rs635299, variant T) and with significant enrichments for H3K9me3 and, unexpectedly, also for H3K27me3, similar to the observed in transgenic mice[13]. Interestingly, 5-Aza-dC treatment was generally inefficient in removing aberrant methylation levels in the gene-edited clones, hence suggesting that it may be more complicated than originally thought to remove abnormal modifications from patients' cells even when combining gene editing with chemical-based approaches.

We next addressed the question of whether differentiation blocks the reversibility of this process while ruling out the possibility of a cell type-specific phenomenon. The findings showed that akin to observations in patient myoblasts, a complete deletion of the CTGs in TOFs (teratoma-derived fibroblasts) originating from DM1-affected hESCs failed to abolish abnormal methylation levels. Furthermore, by generating iPSCs from CTG-deficient affected myoblasts, we showed that de-differentiation can restore, at least in part, the normal hypomethylated status of the gene. Collectively, these results underscore the role of differentiation as a 'point of no return' in the plasticity of this process.

Although we expected complete eraser of abnormal methylation in the repeat-less iPSCs, it was often reduced rather than fully eliminated. The varying levels of residual methylation in the corrected iPSCs could be attributed to the random integration of the lenti-viral OSKM vector, known for its increased reprogramming efficiency but linked to diverse epigenetic profiles in different clones[31]. Additionally, the myoblast cell lines' origin, established through hTERT and CDK4 immortalization, might hinder the reprogramming process[31]. On the other hand, it has been shown that some loci have a strong propensity to be insufficiently or aberrantly reprogrammed, thus providing hotspots for aberrant epigenomic reprogramming[32].

To gain mechanistic insights concerning the plasticity of abnormal methylation in pluripotent cells, we searched for chromatin modifiers that may possibly explain the underlying distinction between undifferentiated and differentiated cells in the context of aberrant methylation maintenance at the DM1 locus. We determined the expression profiles of nearly 150 potentially pertinent genes, encompassing the entire spectrum of chromatin modifiers within the genome. Remarkably, we uncovered substantial changes between these two cellular states- human embryonic stem cells (hESCs) vs. terminally differentiated fibroblasts (TOFs/myoblasts). Notably, the de novo DNA methyltransferases *DNMT3a* and *DNMT3b*, along with the demethylating enzymes *TET1* and *TET2*, emerged as the most significant differentially expressed genes. *DNMT3b* and *TET1* displayed exclusive and particularly high expression levels in undifferentiated hESCs, while *DNMT3a* and *TET2* exhibited a significant increased/decreased expression, respectively.

Given these observations, and the known role of DNMT3b as the primary and most active de novo DNMT enzyme during preimplantation embryo development[33], with a well-established role in shaping methylation patterns within repetitive elements[34–37], we first knocked out *DNMT3b* in the DM1 hESCs. However, the *DNMT3b*-null DM1 hESC clones did not manifest any discernible changes in aberrant methylation levels when compared to unmanipulated matched control (Fig. 4c). One plausible explanation for this could have been insufficient cell proliferation. However, this was highly unlikely given the extensive cell growth of our hESCs in culture. Alternatively, this could have been attributed to functional redundancy between DNMT3b and DNMT3a, as almost all DMRs in the genome (96%) are redundantly targeted by both enzymes, and only lose methylation when both are ablated in hESCs[29]. To address this possibility, we proceeded with the disruption of *DNMT3a* in the background of pre-existing *DNMT3b*-null DM1 hESCs. This involved generating homozygous deletions, resulting in either a complete absence or, more commonly, residual levels of most likely ineffective transcripts due to exon 19 skipping, on the genetic background of DM1 *DNMT3b* KO hESCs. Indeed, we observed a significant decline in abnormal methylation levels in 4 out of the 5 assayed clones, demonstrating reductions that ranged from 18% to 38.5% of the maximum 50% (Fig. 4e). This finding emphasizes the vital role of de novo DNMTs, DNMT3a alone or more likely in combination with DNMT3b, in maintaining abnormal methylation patterns in undifferentiated DM1 hESCs. Moreover, it underscores a critical distinction in the mechanism employed to maintain aberrant methylation patterns between undifferentiated and differentiated cells. This is evident as the abnormal methylation in TOFs and myoblasts remains unaffected following repeat deletion, despite the silencing of *DNMT3b* and the significant reduction in *DNMT3a* levels by differentiation.

Why some DNMT3-double targeted clones are more efficient in removing hypermethylation than others remains unknown. One potential explanation is that de novo methylation by DNMT3a and/or DNMT3b may inconsistently compensate for the inefficient activity of DNMT1 next to the CTG repetitive sequences[38]. Nevertheless, it should be noted that unlike the single *DNMT3b* KOs, these DNMT3-deficient clones exhibited morphological changes and proved incapable of sustaining growth beyond five passages. In contrast to Liao et al.'s previous report, where the deletion of DNMT3 enzymes showed no apparent impact on hESCs[29], our study highlights specific differences. Our investigation revealed complete absence of all forms of *DNMT3b*, while their single/double *DNMT3b* KOs retained the inactive form. Additionally, the sequence and methodology for generating double targeted clones differed, and our analysis involved multiple DNMT3-deficient clones, potentially representing a more severe state compared to the single representative clones in their study[29].

In summary, our study uncovers a fundamental distinction between undifferentiated and differentiated cells in terms of the role played by de novo DNMTs in maintaining abnormal methylation patterns at the DM1 locus in undifferentiated hESCs. This molecular event is dependent on the DNA sequence i.e., the CTG expansion at the 3'-UTR of the *DMPK* gene. This is consistent with the view that the establishment of DNA methylation patterns during preimplantation development is sequence-specific, while their maintenance in the soma is not[39].

We hypothesize that the 3'-end of *DMPK*, which resides within a CpG island and almost overlaps with the *SIX5* promoter, normally remains free of methylation by the high activity of TET1 in pluripotent stem cells[40]. However, as soon as the CTGs expand (>300 triplets), *DMPK* becomes hypermethylated by the counteracting activity of the de novo DNMTs (DNMT3a/b), which are wrongly recruited to the locus (see Fig. 5 for model). This is achieved in a reversible fashion due to the high and opposing activity of the de-methylation enzyme(s). Once the cells differentiate, aberrant methylation becomes fixed in the cells by a shift from de novo (DNMT3) to maintenance (DNMT1) DNMT activity, coinciding with the inability of TETs to remove methylation from CpG island harboring promoters in differentiated cells[41]. This eventually results in permanent propagation of already established methylation patterns and takes place in a mutation-independent fashion.

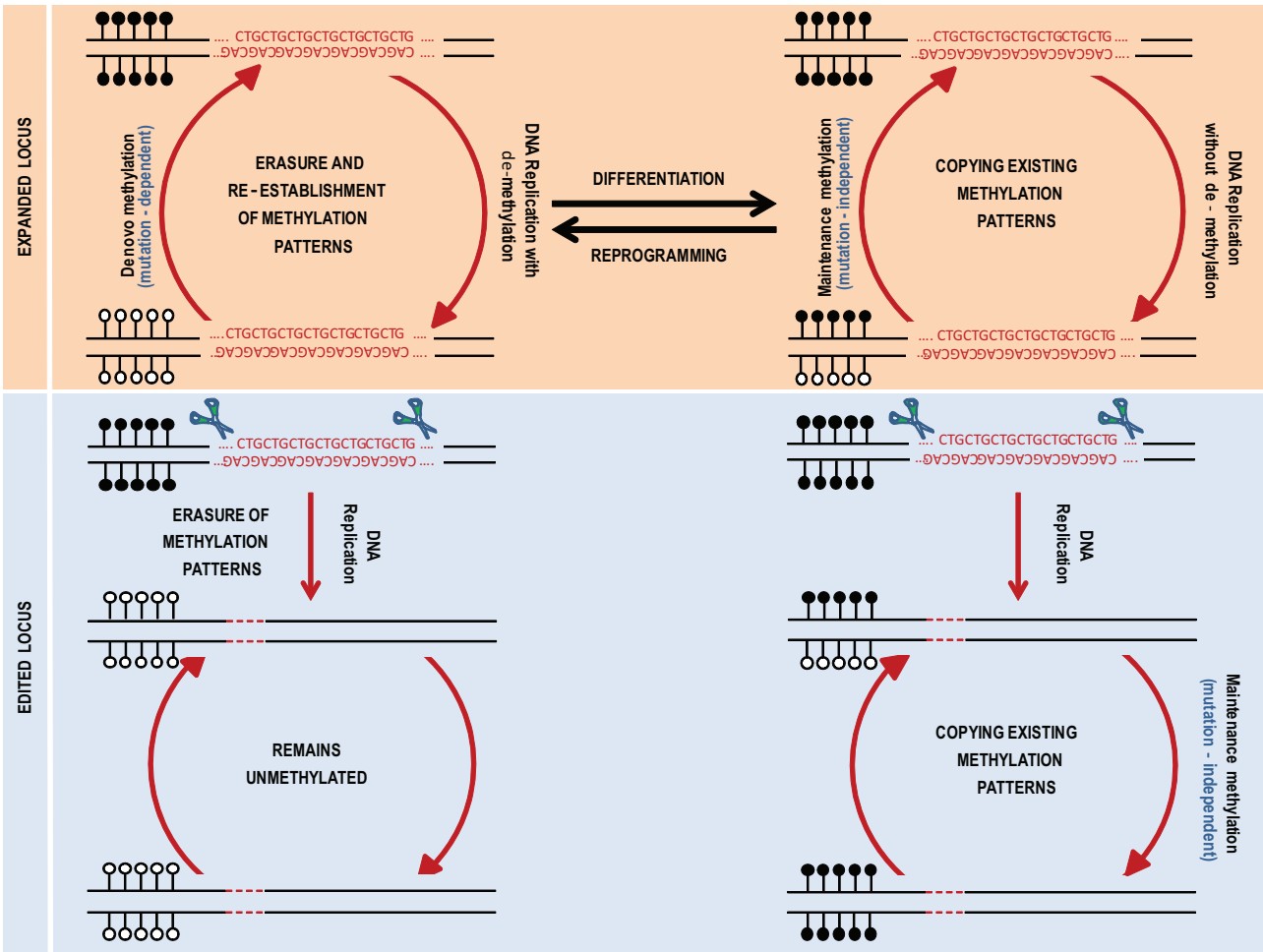

**Fig. 5 | A model for the shift from reversible to irreversible aberrant methylation in DM1 by cell differentiation.** *Top panel*: In DM1 undifferentiated embryonic cells (hESCs), every DNA replication cycle aberrant methylation is repeatedly erased by TET1, and then re-established by de novo DNMTs activity (initiation step) in a way that depends on the presence of the mutation. Once the cells differentiate (skeletal muscle), aberrant methylation patterns remain unchanged by the activity of maintenance DNMTs ("locking up" step), independent of the DNA sequence.

*Bottom panel*: When the CTG expansion is excised from a heavily methylated allele with a large CTG expansion, it results in the loss of aberrant methylation in undifferentiated cells (reversible). This contrasts with the differentiated cells, where excision of the CTGs did not change the epigenetic status of the locus (irreversible). Black lollipops correspond to methylated CpGs and white lollipops represent unmethylated CpGs.

Clearly, the failure to reset the epigenetic status of the DM1 locus simply by correcting the DNA sequence in patients' cells may be relevant and should be taken into consideration when viewing the mutation as a potential therapeutic target for addressing the epigenetic aspects of any disease-causing mutations that coincide with aberrant epigenetic modifications at one or more loci (for the list of conditions see[4]).

## Methods

This research was performed in compliance with the Ethic Committee of Shaare Zedek Medical Center (SZMC) and the National ethic committee (Israel health ministry), and the joint ethics committee (IACUC) of the Hebrew University and SZMC. The Hebrew University is an AAALAC international accredited institute.

### Cell culture

The establishment and use of hESC lines (DM1-affected and wild types) from PGD derived embryos was performed in compliance with the protocols approved by the Ethics Committee of Shaare Zedek Medical Center (IRB 87/07). Embryo donations were carried out under the strict regulation of the National Ethics Committee (Israel Health Ministry) and NIH and ISSCR guidelines. The donated embryos are byproducts

of PGD treatments and would otherwise have been destroyed. Complete separation was maintained between the individual who approached the patients and received informed consent for donation (genetic counselor), the attending physician (IVF gynecologist) and the researcher. While obtaining informed consent, the general aim of the research was explained to the patients. The patients were approached for donation only once and not every IVF cycle, to ensure that there was no connection between the signing of the informed consent form and medical treatment. There was no monetary compensation for the embryo donation.

Human embryonic stem cell (hESC) line derivation and characterization were carried out as described previously[42]. Unaffected (SZ-13 and SZ-RB26) and DM1 affected hESCs (SZ-DM14 carrying a CTG2000 expansion), their gene-edited counterparts and all iPSCs were cultured in hESC media (knockout DMEM (Gibco) supplemented with 8 ng/mL basic fibroblast growth factor (PeproTech)). HEK-293T cells were grown in Dulbecco's modified Eagle's medium, supplemented with 10% fetal bovine serum and 0.5% Penicillin Streptomycin.

DM11 myoblast cell lines, immortalized by overexpression of hTERT and CDK4[43], and their subclones before and after gene editing were cultured as described in[25].

## Reprogramming of myoblasts

To induce the reprogramming of myoblasts, a Doxycycline-inducible OKSM expressing STEMCCA cassette was used together with M2rtTA expressing plasmid at a ratio of 4:1. For the infection process, replication-incompetent lentiviruses containing the STEMCCA cassette or M2rtTA plasmid (10 μg) were packaged with a lentiviral packaging mix (7.5 μg psPAX2 and 2.5 μg pGDM.2) in 10 cm plates of 293 T cells. Lentiviruses were collected 48, 60, and 72 hrs after transfection. The supernatants were filtered through a 0.45 μm filter, supplemented with 8 μg/ml of polybrene (Sigma), and then used to infect the myoblasts. Six hours after the third infection, the medium was replaced with fresh DMEM containing 10% FBS. Eighteen hours later, 2 μg/ml Dox was added to the plates. 10% FBS DMEM medium with Dox was replaced every other day for 7 days, then replaced by a 1:1 mix of 10% FBS DMEM and hESC medium (knockout DMEM supplemented with 14% knockout serum (Gibco), 0.1 mM β-mercaptoethanol, 2 mM L-glutamine, 1% non-essential amino acids) with 2 μg/ml Dox until days 8–14. At days 15–20 of reprogramming, the cells were grown in a full hESC medium supplemented with Dox. On day 21, Dox was removed from hESC medium and 4 ng/ml of hFGF2 (Peprotech) was added. 5–10 days after Dox removal, the plates were screened for primary iPSC colonies. 12 single iPSC colonies were manually picked using a Pasteur pipette and plated in separate wells in a 6-well plate on feeder cells. Wells were monitored and the medium was replaced every day for 3–5 passages, until stable colonies developed.

## Teratoma production

Teratomas were induced by subcutaneous injection of $1–5 \times 10^6$ undifferentiated hESCs into SCID/beige mice. The animals were kept in a controlled environment, meeting specific pathogen-free (SPF) standards, within the animal facility at the Life Sciences Institute—the Hebrew University of Jerusalem. The housing conditions, including 12 h light and dark cycles, temperature maintained at 22 °C ± 2 °C, and humidity ranging from 30% to 70%, were meticulously regulated by the Hebrew University Authority for Biological and Biomedical Models (ABBM). The study protocol (IACUC# NS-21-16595-4) was ethically approved by the joint ethics committee (IACUC) of the Hebrew University and Shaare Zedek Medical Center to ensure the welfare of the animals. Furthermore, the Hebrew University holds accreditation from the AAALAC International (#1285), reinforcing its commitment to the highest standards of animal care and research ethics.

Tumors were isolated 5–8 weeks after the injection. Tumor-derived fibroblast-like cultures were established by manual cutting following trypsin dissociation, and subsequent plating on gelatin-coated dishes in DMEM supplemented with 10% FCS, 50 units/ml penicillin and 50 mg/ml streptomycin.

## Gene editing by transfection and electroporation

Undifferentiated hESCs were grown in feeder-free culture conditions using defined Essential 8 medium combined with vitronectin-(Gibco)-coated surfaces. One day before transfection, $1.25 \times 10^5$ hESCs were seeded per 6 wells with 10 μM ROCK inhibitor Y-27632 (PeproTech). To generate clonal lines, hESCs were co-transfected with two pSpCas9(BB)−2A-Puro (PX459) V2.0 plasmids, each expressing either the upstream or downstream gRNAs, as detailed in Table S2. 24 h post transfection, 0.2 μg/ml of puromycin was added to the media for 48 h. Feeder cells (mitomycin C-treated mouse embryonic fibroblasts) were laid on the transfected cells to allow expansion of the clonal lines. Single cell puro-resistant clones were isolated manually and re-plated onto the feeder cells.

For the teratoma-derived fibroblasts, targeting vectors were introduced into the cells by electroporation ($1 \times 10^6$ cells) with the Neon transfection system (Thermofisher) according to the following parameters: 1650V, 10 ms, 3pulse, tip type 100 μl. Immediately after electroporation, the cells were plated on gelatin-coated dishes. 24 h post transfection, the cells were treated with 0.2 μg/ml of puromycin for 48 h. Puro-resistant cells were further propagated and collected for DNA extraction after trypsinization.

## Cloning of gRNAs

Guide RNAs (gRNAs) were designed to target the upstream and downstream regions flanking the CTG repeat using the Zhang lab CRISPR design tool (crispr.mit.edu). Both gRNAs were cloned into the pSpCas9(BB)−2A-GFP (PX458) and pSpCas9(BB)−2A-Puro (PX459) V2.0 from Feng Zhang (Addgene #48138, #62988 respectively) according to the protocol described in[44]. In brief, complementary DNA oligomers containing the gRNA sequence and a BbsI restriction site were annealed and ligated into PX458 or PX459 with T4 DNA ligase (New England Biolabs). Insertion of the target sequence into the plasmid was verified by DNA sequencing using the U6 Forward primer (5′- GAGGGCCTATTTCCCATGATTCC-3′).

## PCR screening for edited alleles

To validate the targeted deletion of the CTG repeat, genomic DNA was extracted from a small number of cells of each clone and analyzed by Gene Scan analysis. In brief, one colony from each clone was manually isolated and lysed in an alkaline lysis reagent (25 mM NaOH, 0.2 mM EDTA) at 95° C and neutralized by a 20 mM Tris buffer (pH 7.4). Lysed cell samples were directly used in PCR and the products were analyzed by Gene Scan with an ABI 3130 DNA Analyzer to determine the size of the amplified product (5′FAM primer: 5′-CAGCTCCAGTCCTGTGATCC-3′ and 3′ primer: 5′-CACTTTGCGAAC-CAACGATA-3′) and further validated for the excision of the repeat tract by Sanger sequencing. To distinguish between homozygous null and heterozygous deletions, PCR was carried out across the breakpoints. A similar approach was used to validate bi-allelic deletion for *DNMT3b* and *DNMT3a* (Fig. S7b and S7d, respectively), employing the primers outlined in Table S2.

## Assessment of off-target effects

To assess the potential off-target effects of CRISPR/Cas9 plasmids with 7gRNA or 44gRNA elsewhere in the genome, we utilized the off-target tool from the WTSI Genome Editing (WGE) website. Potential off-target sites were identified based on similarity to the selected gRNA sequences. Sanger sequencing of these loci was carried out to confirm that no indels were induced by gene editing, as determined by this assay (Fig. S2b).

## Southern blot analysis

Genomic DNA samples (10–21 μg) from candidate clones and unmanipulated controls were digested with SacI and HindIII (Fermentas) restriction endonucleases, separated on 0.8% agarose gels, blotted onto Hybond N+ membranes (Amersham), and hybridized with a PCR Dig-labeled pDM576 probe (primers: 5′-GCTAGGAAGCAGCCAATGAC-3′ and 5′-CATTCCCGGCTACAAGGAC-3′), as previously described[14]. Detection of DNA fragments was carried out using CDP-Star Chemiluminescent Substrate for Alkaline Phosphatase (Roche).

## Locus-specific bisulfite colony sequencing

Genomic DNA (2 μg) was modified by bisulfite treatment (EZ DNA methylation kit, Zymo Research) and amplified by either FastStart DNA polymerase (Roche) or Takara Epitaq HS (Takara Bio). Amplified products using universal bisulfite converted primers that equally detected methylated and unmethylated alleles were TA-cloned, transformed and single colonies were analyzed for CpG methylation by direct sequencing (ABI 3130) using T7 and Sp6 primers. Molecules (each represented by a row) are considered methylated if at least 50% of the CpGs (13/26 and 8/17 potential sites in regions E and F, respectively) remain unaltered after bisulfite conversion.

## Locus-specific bisulfite deep-sequencing

Genomic DNA (2 μg) was modified by bisulfite treatment (EZ DNA methylation kit, Zymo Research) and amplified by FastStart DNA polymerase (Roche). PCR was carried out using 5′ primer: 5′-TCGT CGGCAGCGTCAGATGTGTATAAGAGACAG<u>TGGTTGTGGGTTAGTGTT</u> −3′, and 3′ primer: 5′-GTCTCGTGGGCTC<u>GGAGATGTGTATAAGAGAC AGCCCAACCAACCTACAACTATTAT</u>−3′ (*DMPK*-specific sequences are underlined; adapters for library preparation are at the 5′ end of each primer). Amplified products were quality controlled for size using the D1000 ScreenTape kit (Agilent Technologies) and for concentration using the Qubit® DNA HS Assay kit (catalog #32854; Invitrogen). Subsequently, the PCR products were bead purified with Agencourt Ampure XP beads (Beckman Coulter) according to the manufacturer's protocol. Next, amplicons were subjected to a second PrimMax Takara PCR reaction (1st purified PCR DNA (7.5 μl), 2.5 μl N7XX primer (nextera barcode 1), 2.5 ul S5XX primer (nextera barcode 2), 12.5 μl 2x Primstar ReadyMix). PCR program: 98 °C for 1 min followed by 8 cycles of 98 °C for 10 s, 55 °C for 10 s, 72 °C for 30 s, then 72 °C for 5 min and finally held at 10 °C. The 2nd PCR was bead purified, quality controlled using the D1000 ScreenTape kit and the Qubit® DNA HS Assay kit, and all resultant libraries were normalized and pooled at a 10 nM concentration. Denaturation and dilution of the sequencing pool were performed according to the standard Illumina protocol. Ultimately, 1.5 pmole of pool (combined with 40% spiked-in) was loaded onto a NextSeq 500 Mid-Output Kit (150 cycles) cartridge (catalog #FC-102-1001; Illumina, San Diego, CA) for high throughput sequencing on a NextSeq 500 instrument (Illumina), with 150 cycle, single-read run conditions.

Raw sequence reads were mapped to the human genome (build GRCh37) using Bismark (version 0.19.1)[45]. Methylation calls were extracted after duplicate sequences were removed. Data visualization and analysis were performed using custom R and Java scripts. CpG methylation was calculated as the average methylation at each CpG position, and non-CpG methylation was extracted from the Bismark reports. The raw (fastq files) and analyzed (text files) data related to deep-sequencing for *DMPK* methylation following bisulfite treatment were deposited at the NCBI GEO under accession number GSE128901.

## RT-PCR/RT-qPCR

Total RNA was isolated from cells by TRI reagent extraction. RNA (1 μg) was reverse transcribed (Multi Scribe RT, ABI) with random hexamer primers. Amplification was performed under PCR conditions and primers as detailed in Table S2.

Real-time PCR was performed using Power SYBR Green Master Mix (ABI) on an ABI 7900HT instrument using primers listed in Table S2.

## Droplet RT-PCR

The expression level of *DMPK* and *SIX5* mRNA was measured by the QX200 Droplet Digital PCR (ddPCRTM). Total RNA was isolated from the cells by TRI reagent, and then 1–2 microgram RNA was reverse-transcribed by random hexamer priming and multi scribe reverse transcriptase (ABI). ddPCR was carried out using BioRad ddPCR Supermix (Cat#1863023) according to the manufacturer's instructions. The TaqMan probes (Applied Biosystems) were *SIX5* (HS1650774), *DMPK* (HS01094329) and *GUS* (HS99999908), as a housekeeping gene for normalization. Droplet reading and analysis were performed using QuantaSoft 3.0 software.

## RNA-seq

Total RNA was extracted from cell pellets using NucleoSpin RNA Plus kit (Macherey-Nagel, Cat # 740984.50) following the manufacturer's instructions. RNA was quantitated by NanoDrop 200 (ThermoFisher). RNA libraries were prepared for sequencing using TruSeq RNA library Prep Kit (Illumina). For transcriptome analysis, RNA-seq reads were trimmed for adapter\low quality sequences using Trimmomatics (v0.39), followed by alignment to GRCh38 reference genome using STAR package. Count tables were established by Subreads' *feature-counts* function. Normalization and differential expression analysis were performed by using the edgeR package. The raw (fastq files) and analyzed (text files) data related to the RNA-seq experiments were deposited at the NCBI GEO under accession number GSE128901. Volcano plots were generated by VolcaNoseR, a web application for creating, exploring, labeling and sharing volcano plots−doi: 10.1038/s41598-020-76603-3. The raw (fastq files) and analyzed (text files) data related to all RNA deep-sequencing experiments were deposited at the NCBI GEO under accession number GSE128901.

## Western blot

Cell pellets were collected in lysis buffer containing 20 mM Tris-HCl (pH 8), 100 mM NaCl, 5 mM EDTA, 1% Triton X-100 in PBS, and a protease inhibitor. Resulting lysates were then electrophoretically resolved on a 4–20% polyacrylamide gel (Bio-Rad) and transferred to nitrocellulose membranes using the wet transfer method. Subsequently, the membranes were blocked with PBST (3% skim milk powder) for 30 min at room temperature (RT) and then incubated in blocking buffer overnight at 4 °C with a monoclonal antibody against DNMT3b protein (Santa Cruz Biotechnology sc-376043, 1:500 dilution) or GAPDH (Abcam ab8245, 1:2,000 dilution) as a loading control. The DNMT3b monoclonal antibody targets amino acids 1–230, yielding a 97 kDa product. After an incubation step with a goat anti-mouse horseradish peroxidase (HRP)-conjugated antibody (R&D Systems HAF007, 1:5000 dilution) for 1 h at RT, detection was carried out using an enhanced chemiluminescence detection kit according to the manufacturer's protocol (Biological Industries).

## Chromatin immunoprecipitation

Chromatin immunoprecipitation was performed as described[46]. In brief, cells were harvested and then fixed, quenched, and washed in 50-ml tubes. Sonication was carried out using a Vibra Cell VCX130 with a 3 mm microtip and 30% amplitude in 4 cycles of 4 min for hESCs and 25 cycles of 25 min for myoblast cells. Immunoprecipitation was performed using an anti-H3K9me3 (Abcam Ab8898), anti-H3K27me3 (Abcam Ab6002) and an anti-CTCF (Diagenode C15410210) antibodies. Real-time PCR was carried out on an ABI 7900HT instrument (primers are listed in Table S2). −ΔCt values were normalized according to the negative control to account for differences in precipitation efficiency. *APRT* served as a negative control for H3K9me3, H3K27me3 and CTCF in all experiments. *HOXA9* served as a positive control in hESCs for both histone modifications, whereas *MYOGENIN* served as a positive control in myoblasts exclusively for H3K9me3. *FXN* served as a positive control in hESCs for CTCF.

## Statistics and reproducibility

Precise *p*-values for all experiments (Figs. 1b, c, 2b, c, 4e, S3c and Sd) are provided in Table S4.

## Reporting summary

Further information on research design is available in the Nature Portfolio Reporting Summary linked to this article.

# Data availability

The data sets pertaining to locus-specific bisulfite DNA deep-sequencing and RNA deep-sequencing generated in this study have been deposited in the NCBI GEO database under accession number GSE128901. Source data for all ChIP (H3K9me3, H3K27me3 and CTCF experiments), RT-qPCR (*DNMT3a*) and RT-dPCR (*DMPK and SIX5*) experiments are provided as a Source Data files. Source data are provided with this paper.

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

## Acknowledgements

We would like to thank Prof. Bé Wieringa for critical reading of the manuscript. The immortalized myoblast cell lines ASA308DM1 (DM11cl5, 2600CTG) and AB1079 (LHCN-M2) were kindly provided by the platform for immortalization of human cells at the Institut de Myologie, Bâtiment Babinski, GH Pitié-Salpétrière. This work was partly supported by the Ministry of Science and Technology (MOST) Israel (3-17372, to R.E.), Israel Science Foundation (244/23 to R.E.), a donation from the Abrasba Foundation (Gindi family, to R.E), the Austrian Science Fund (FWF I 4768-B to S.K.), ZonMw (TOP grant NL91212009, to Bé Wieringa and DGW) and the Prinses Beatrix Spierfonds (grant number W.OR16-09, to DGW).

## Author contributions

T.H., M.A.D, S.J., S.Y.D., U.A., R.S.G., S.M., E.B., S.B.S., Y.B. and S.E.L. conducted the experiments and analyzed the data; F.Z. conducted the bioinformatic analysis, Y.D., S.K., W.V.D.B., D.G.W. and V.M. contributed reagents/materials, R.E. designed the experiments, R.E. and D.G.W. wrote the paper, Y.B. and S.K. were responsible for a critical reading of the manuscript.

## Competing interests

The authors declare no competing interests.
