## [Peer Review File · Nature Communications]

Differentiation shifts from a reversible to an irreversible heterochromatin state at the DM1 locusREVIEWER COMMENTS

Reviewer #1 (Remarks to the Author):

In this paper, the authors examine the role of expanded CTG repeats on the maintenance of heterochromatin during differentiation using genome-edited DM1 cell models. The authors demonstrate that abnormal methylation and H3K9me3 enrichments of the DM1 locus are abolished by CTG repeat excision in undifferentiated DM1 hESCs, but CTG excision in affected myoblasts cannot restore the normal epigenetic status. They also show the shift from a reversible to an irreversible heterochromatin state by hESC differentiation. While some potentially interesting results have been observed, the primary weakness of this paper is the lack of mechanistic data. The authors have not addressed the question of how differentiation participates in the maintenance of repressive epigenetic modifications. The problem with this study is that it is rather inferential and observational. I feel that a substantial revision would be required prior to publication.

1. The authors have analyzed only methylation in region E of the DM1 locus. The methylation analysis in region F should be done because the region contains a putative CTCF binding site. The CTCF-binding properties of repeat-cleavage-induced changes in methylation in region F can be verified by ChIP, which the authors used in their previous paper (Stem Cell Reports. 2015 Aug 11;5(2):221-31.). Changes in SIX5 and DMPK expression should also be examined, as was done in the same paper.

2. The authors analyzed only H3K9me3 by the ChIP study. Since there are many other repressive (and active) chromatin markers, ChIP with these other markers should also be considered.

3. Lines 178-181: the authors state "To detect aberrant methylation upstream to the repeat, the SacI-HindIII fragments were further processed by parallel restriction with MspI (+MspI) and its methylation-sensitive isoschizomer, HpaII (+HpaII)." Where is the result? There is no indication of +MspI and +HpaII in Figure S2.

4. Lines 301-303: how can the author prove that clone 9 and clone 29 are different clones?

5. Lines 326-328: "Locus-specific deep-sequencing in an overlapping region (15 CpGs, -759 bp to -631 bp relative to the repeat) in the parental cell line and Δ/Δ clones validated these results (accession number GSE128904)." The authors should provide a better explanation for this validation from the deep-sequencing data, for example, showing the result in Figure 1.

6. Figure 2A: there are several non-methylated CpGs in these myoblast clones, and it doesn't appear to be 100% methylated.

7. Lines 376-377: "Indeed, we found reduced methylation levels at the DMR in one out of two gene-edited drug-treated myoblast clones (Figure 2C)," Why does only one clone show a reduced methylation level after the treatment with 5-AZA-dC? The authors should provide a clearer explanation for the difference observed between M4 and M6.

8. Line 412: "Analysis of DNA methylation levels in the resulting iPSCs was carried out as described for the gene-edited myoblasts by utilizing a non-CpG informative SNP within the DMR (rs635299, see also (12))." The SNP (rs635299) was referred to reference #14 in line 358. Which is correct?

9. Lines 412-414: "This procedure confirmed that reprogramming the gene-edited myoblasts reduced abnormal methylation levels to 93%-0% (Figures 3B and S4B)." The methylation percentage is highly variable among the repeat-less isogenic iPSC clones. The claims in lines 423-427, "As expected, the findings showed an opposite relationship between the extent of residual methylation and TET1 mRNA levels...", are not convincing because the authors draw this conclusion based upon only IPS3.

10. Lines 477-479: "We found an inverse relationship between the extent of residual hypermethylation and the levels of TET1 mRNA in the gene-manipulated iPSCs." It is difficult to conclude this from Figure 3B.

11. Lines 500-509: "We hypothesize that the 3'-end of DMPK, which resides within a CpG island and next to ..." This part of the manuscript is purely speculative. The authors must directly test their hypothesis in their cell models.

12. Figure 3B: what do the error bars indicate?

13. Figure S2B: some of the sequences of CL29 are not aligned accordingly.

14. The authors should clearly mention that they used immortalized myoblasts in the Materials and Methods rather than Acknowledgement.

Reviewer #2 (Remarks to the Author):

The manuscript entitled "Differentiation shifts from a reversible to an irreversible heterochromatin state at the DM1 locus" by Handal et al. investigates the deposition of DNA methylation and heterochromatin associated H3K9me3 at the DM1 locus with an expanded trinucleotide repeat. The authors use pluripotent stem cells from patients with an expanded repeat and show that this leads to methylation and H3K9me3. CRISPR mediated excision of the trinucleotide repeat reverses both modifications. This is in line with a potential to reprogram heterochromatin in human pluripotent stem cells and contrasts earlier findings in patient derived myoblasts, which are unable to reverse methylation upon excision of a similar region. The authors use these myoblasts from earlier work to show that reprogramming of myoblasts can reverse methylation only if the expanded repeat has been removed. However, it appears that not in all instances repeat removal is sufficient for demethylation (M6 in fig 3) - which is surprising as the authors result would have suggested if a pluripotent state is reached removal of the expanded repeat would cause reversion of heterochromatin. Finally, the authors derive teratoma in immune compromised mice and establish fibroblastoid like cell lines with the expanded trinucleotide repeat. CRISPR mediated excision of the expanded repeat does not lead to reversion of DNA methylation consistent with the view that differentiated cells lack the ability to reverse heterochromatin. Overall this is a well performed study with a large amount of data that will be of interest for readers in chromatin biology and trinucleotide repeat diseases. However, there are several aspects that would need to be further addressed before a conclusions can be drawn.

Specific points:

1. The introduction informs that repeat expansion leads to a autosomal dominant disease allele. It is not clear how DNA methylation and potentially silencing of the downstream gene are related to the disease phenotype. Is heterochromatinization causing a haploinsufficiency or acting in synergy with a pathogenic triplet expanded RNAs? Importantly, would a disease phenotype be expected with the expanded repeat is removed irrespective if heterochromatin remains? This could have implications in the relevance of the heterochromatinization aspect for treatment. If persisting heterochromatin were not causing a phenotype after repeat deletion potentially engineered myoblasts could be considered for treatment. It would be good if this could be clearly state in the introduction or discussion to inform the reader.

2. The authors use previously established myoblast lines where an expanded repeat has been deleted and persistence of heterochromatin has been observed. Importantly, they explore if heterochromatin can be reversed by iPSC reprogramming. Although this experiment has a strong expectation that in a pluripotent cell elimination of the repeat should allow reprogramming, the outcome in figure 3 and text line 414 "level to 93%-0%" provide a mixed view. At first glance the 93%-0% appears as a type but at closer inspection it appears that the authors are serious about these numbers. Since 93% and 0% are nearly and exactly at the end of the possible scale of

outcomes one would have expected a different discussion. The data clearly show that whereas clone M4 can reverse heterochromatin, clone M6 does not. An explanation such as M6 was not fully reprogrammed is needed to resolve this issue. In my mind either the authors hypothesis that pluripotent cells always reverse DNA methylation after repeat excision or the myoblasts used are problematic.

3. The authors argue for a shift from reversible DNA methylation in early embryonic (pluripotent) cells to non-reversible DNA methylation in more differentiated cells (MEFs, myoblasts) based on their observations after CRISPS mediated repeat excision. This is very reasonable and in line with findings in other systems. However, there could also be an influence of the culture conditions. It is known that cells lines in traditional cell culture media a prone to get non-physiological methylation and this could contribute at DM1. To investigate if other regions in the genome might have become aberrantly methylated due to culture conditions independent of the DM1 trinucleotide expansion, control CpG island methylation need to be determined of genes that are normally not methylated but have been shown to become methylated in cultured cell lines. In addition, a comment on the presence of media components that are known to enhance demethylation in the different (iPSC, MEF, and myoblast) media should be considered in the discussion.

Reviewer #1 (Remarks to the Author):

1. The authors have analyzed only methylation in region E of the DM1 locus. The methylation analysis in region F should be done because the region contains a putative CTCF binding site. The CTCF-binding properties of repeat-cleavage-induced changes in methylation in region F can be verified by ChIP, which the authors used in their previous paper (Stem Cell Reports. 2015 Aug 11;5(2):221-31.). Changes in *SIX5* and *DMPK* expression should also be examined, as was done in the same paper.

Response: We determined methylation levels by bisulfite colony sequencing in region F and validated the binding of CTCF in an overlapping region by ChIP analysis in all hESC genotypes including wild type, DM1 and Δ/Δ clones, as requested (Fig S3b and S3c). Although hypermethylation in region F is exclusive to expanded alleles (Figure S3b), CTCF enrichment levels are statistically indistinguishable across all cell types when compared to DM1 hESCs (DM1) (Figure S3c). This is because despite the exclusive binding of CTCF to unmethylated alleles (see Fig S5D and S5E in Yanovsky-Dagan et al. 2015), our ChIP assay was not sensitive enough to detect a 2-fold difference in enrichment levels (comparing the wild type to DM1 hESCs). Nevertheless, because gene editing does not appear to impede CTCF binding (as evidenced by the enrichment levels in Δ/Δ clones), it is very likely that the loss of heterochromatin by gene editing coincided with the recovery of CTCF occupancy on the mutant allele, as expected.

We now state this in the text as follows:

"It has been suggested that aberrant methylation patterns upstream of the CTGs might alter the chromatin structure through the loss of CTCF binding. Consistent with this claim, we determined methylation levels by bisulfite colony sequencing immediately upstream to the repeat (region F) and validated the binding of CTCF in an overlapping region (CTCF binding site I, CTCFI) by ChIP analysis in wild type, DM1 and Δ/Δ hESCs (Figure S3b and S3c). Although we were unable to distinguish between the wild type and the affected hESCs in terms of the extent of enrichment using this ChIP assay, we found that hypermethylation at the CTCF binding site is exclusive to DM1 hESCs (Fig S3b). The known role of methylation in abolishing CTCF binding at this site in hESCs (Fig S5D and S5E in (14)), along with the fact that gene editing did not seem to interrupt CTCF binding in the immediate proximity of the repeat (as evidenced by the enrichments in Δ/Δ clones), imply that the loss of heterochromatin by gene editing most likely restored CTCF occupancy on the mutant allele." (Pages 14-15, Lines 364-376).

With respect to the *DMPK* and *SIX5* expression levels, we now show by RT-ddPCR that the wild type and the affected DM1 hESCs do not differ significantly in terms of their total *DMPK* and *SIX5* mRNA levels (Fig S3d). Nor do they change (*SIX5* mRNA) or consistently trend (*DMPK* mRNA) as a result of CTG excision when comparing DM1 unmanipulated vs. gene-edited hESCs (Figure S3d). These findings, which are corroborated by our published data reporting no change in mRNA levels between DM1 unmanipulated and gene-edited myoblasts (see Van Agtmaal et al. 2017), strongly suggest that local gene expression is

unaffected by abnormal methylation at the DM1 locus, at least not in undifferentiated embryonic cells, as was previously suggested (Yanovsky-Dagan et al. 2015).

We now state this in the text:

"It has been claimed that the change in chromatin structure may alter local gene transcription at the DM1 locus (14, 31, 32). Therefore, we assessed the total mRNA levels of *DMPK* and *SIX5* in the wild type, DM1-affected and DM1- Δ/Δ hESCs by RT-ddPCR. The results showed that the wild type and affected DM1 hESCs did not significantly differ for total *DMPK* and *SIX5* mRNA levels (Figure S3d). Nor could we find a significant change in *SIX5* mRNA levels or a consistent trend in *DMPK* expression, when comparing DM1 unmanipulated vs. gene-edited hESCs (Figure S3d). Thus, in conjunction with our published data reporting no change in *DMPK* and *SIX5* mRNA expression levels in DM1 unmanipulated as compared to gene-edited myoblasts (see (27)), this strongly suggests that local gene expression is unaffected by abnormal methylation at the DM1 locus, at least not in embryonic stem cells and patients' myoblasts." (Page 15, Lines 377-387).

We refer to the effect of repeat excision in hESCs on CTCF binding and local gene transcription in the Discussion as follows:

"In addition, we provide indirect evidence that the loss of repressive epigenetic marks is coupled with the rescue of CTCF binding near the repeat. Nevertheless, we could not find evidence for a significant change in *SIX5* mRNA levels or consistent trends in *DMPK* mRNA, before or after CTG excision. This was also true between wild type and DM1 affected hESCs, and contrasts with previous suggestions regarding the role for hypermethylation in regulating local gene transcription in DM1 (14, 31, 32)." (Page 19, Lines 489-494).

2. The authors analyzed only H3K9me3 by the ChIP study. Since there are many other repressive (and active) chromatin markers, ChIP with these other markers should also be considered.

Response: We thank the reviewer for insisting that we extend the analysis to other histone modifications. Strikingly, we indeed found a fundamental difference between hESCs and myoblasts. Whereas in undifferentiated hESCs there was no enrichment for H3K27me3 in any of the cell lines/clones, in the myoblasts, DNA hypermethylation and H3K9me3 enrichment (representing constitutive heterochromatin) were coupled with the gain of H3K27me3. In light of these findings, we suggest that the locking mechanism for heterochromatinization by cell differentiation may be related to the general deposition of H3K27me3, in addition to other known modes of DNA methylation and H3K9me3 acquisition.

Note that given the background of the wild type allele, we chose to limit the analysis to repressive chromatin marks since based on our experience, our ChIP assays are not sensitive enough to identify 2-fold differences in enrichment levels.

We now state this in the text as follows:

"To explore whether hypomethylation is coupled with the loss of heterochromatin, we analyzed the enrichment of two repressive histone modifications: H3K9me3 (representing constitutive heterochromatin) and H3K27me3 (representing facultative heterochromatin) immediately upstream to the repeat, by chromatin immunoprecipitation (ChIP). While H3K9me3 was exclusively enriched in the unmanipulated DM1 hESCs (Figure 1b), neither of the hESCs lines/clones were enriched for H3K27me3 (Figure 1c). Hence, H3K9me3

deposition, but not H3K27me3, is tightly correlated with CTG array size and the gain of aberrant methylation in hESCs. These results indicate that repeat removal in mutant hESCs alters the epigenetic status of the locus in a way that prevents constitutive heterochromatin from being re-established near the repeat." (Page 14, Lines 350-360).

"To explore whether hypermethylation was coordinated with heterochromatin, we confirmed significant enrichment for H3K9me3 by ChIP analysis in all of the affected (successfully and unsuccessfully gene-edited) myoblast clones (Figure 2b). Strikingly however, when we monitored for H3K27me3, we observed significant enrichment levels in all the examined cell clones, before and after editing. This suggests that H3K27me3 is elicited with heterochromatinization in a way that depends on differentiation into myoblasts." (Page 16, Lines 414-420).

"Strikingly though, when our experiment was replicated in DM1-affected myoblasts (CTG2600), gene correction failed to restore the epigenetic status of the locus despite many population doublings in culture. This was exhibited, without exceptions, by the methylation levels of 100% in all the tested molecules on the background of the mutant allele (rs635299, variant T) and with significant enrichments for H3K9me3 and, unexpectedly, also for H3K27me3. Our findings on the double marking of H3K9me3 with H3K27me3 in affected myoblasts coincide with results reported by Brouwer and colleagues (13), which showed differential enrichments between unmethylated (H3K27me3 only) vs. hypermethylated alleles (H3K9me3 combined with H3K27me3) in the hearts of transgenic mice carrying the human DM1 locus with different repeat lengths." (Page 19, Lines 498-508).

"It would be crucial to explore the difference(s) in chromatin states between undifferentiated pluripotent vs. differentiated cells (such as differences in composition, nuclear position, or binding of trans-acting factors) to better account for the discrepancy in the reversibility of this process. One potential factor may be the repressive modification H3K27me3, which is a key marker of facultative heterochromatin and is not essential for gene repression in hESCs (39), but is differentially enriched in patients' myoblasts. The potential involvement of H3K27me3 in the locking mechanism of heterochromatin by differentiation should be further explored." (Page 21, Lines 564-571).

3. Lines 178-181: the authors state "To detect aberrant methylation upstream to the repeat, the SacI-HindIII fragments were further processed by parallel restriction with MspI (+MspI) and its methylation-sensitive isoschizomer, HpaII (+HpaII)." Where is the result? There is no indication of +MspI and +HpaII in Figure S2.

Response: We apologize for this careless (copy-paste) mistake. We removed this sentence from the text given that the methylation analysis did not rely on methylation-sensitive Southern blot assay.

4. Lines 301-303: how can the author prove that clone 9 and clone 29 are different clones?

Response: We thank the reviewer for raising this important issue and for drawing our attention to an overlooked 2 bp deletion in one of the alleles in CL29. After careful examination of the Gene Scan results, we noticed a slight widening of the peak in CL29 relative to CL9. This prompted us to re-sequence the PCR products across the junction in both clones, which revealed a 2 bp deletion in one of the alleles of CL29. To further substantiate this finding, we cloned the PCR products into a T-vector and sequenced 13 molecules, 7 of which revealed a 2 bp deletion (CG) next to the breakpoint whereas the remaining molecules indicated perfect repair (data not shown). This contrasts with CL9, where both alleles were accurately repaired. Altogether, by re-sequencing the site of targeting and flanking regions, we provided evidence that CL9 and CL29 are definitively different clones.

The revised text now states:

“While in one clone the sequence was perfectly repaired with no indels in the normal and expanded alleles (CL9), in the other clone (CL29), one allele was perfectly repaired but the second allele had a 2 bp deletion at the junction, thus providing evidence that CL9 and CL29 are definitively different clones in terms of their origin.” (Page 13, Lines 314-317).

5. Lines 326-328: “Locus-specific deep-sequencing in an overlapping region (15 CpGs, -759 bp to -631 bp relative to the repeat) in the parental cell line and Δ/Δ clones validated these results (accession number GSE128904).” The authors should provide a better explanation for this validation from the deep-sequencing data, for example, showing the result in Figure 1.

Response: We now present the results from the deep-sequencing as a heatmap in Figure S3a (instead of Figure 1, as suggested) and state that:

“Locus-specific bisulfite deep-sequencing in an overlapping region (15 CpGs, -759 bp to -631 bp relative to the repeat) in the parental cell line and Δ/Δ clones was used to unequivocally show that methylation was not preserved in any of the successfully edited molecules (see heatmap in Figure S3a, accession number GSE128904).” (Pages 13-14, Lines 341-344).

6. Figure 2a: there are several non-methylated CpGs in these myoblast clones, and it doesn't appear to be 100% methylated.

Response: We routinely determine methylation levels through bisulfite colony sequencing by counting the relative abundance of methylated molecules (represented by rows) rather than by counting individual CpG sites (lollipop). We consider molecules to be methylated if at least 50% of the CpG sites (13/26 potential sites in region E and 8/17 in region F) remain unaltered after bisulfite conversion.

We now state this in EXPERIMENTAL PROCEDURES under “Locus-Specific Bisulfite Sequencing” that: “Molecules (each represented by a row) are considered methylated if at least 50% of the CpGs (13/26 and 8/17 potential sites in regions E and F, respectively) remain unaltered after bisulfite conversion.” (Page 8, Lines 189-192)

7. Lines 376-377: “Indeed, we found reduced methylation levels at the DMR in one out of two gene-edited drug-treated myoblast clones (Figure 2c),” Why does only one clone show a reduced methylation level after the

treatment with 5-AZA-dC? The authors should provide a clearer explanation for the difference observed between M4 and M6.

Response: We extended the 5-Aza-dC experiment to two other gene-edited myoblast clones (altogether 4 different clones, M4-M7). In 3 out of the 4 edited clones, drug treatment was ineffective in reducing methylation levels against the background of the mutant allele (variant T). None of the methylation levels changed in the control unmanipulated DM1 myoblasts (M1). Thus, in the majority of the cases (3/4 edited clones) 5-Aza-dC treatment was unable to restore the normal epigenetic status of the locus against the background of the mutant allele, despite the excision of the repeats.

We now state that:

“Finally, to explore whether persistent methylation in the gene-edited myoblasts could be removed by treatment with a demethylating agent, we monitored for aberrant methylation levels in four different edited clones after a 3-day 5-Aza-dC treatment (5 μ M). Strikingly, in three out of the four tested clones, the methylation levels remained unchanged at the DMR (100% against the background of the mutant allele, Figure S4). In the one remaining clone, the methylation level went from 88% to 55% (M4). None of the methylation levels decreased in the parental DM1 myoblasts as a result of drug treatment (M1). Thus, in most cases (3/4 edited clones), 5-Aza-dC treatment could not restore the normal epigenetic status of the locus against the background of the mutant allele, despite the excision of the repeat.” (Pages 16-17; Lines 423-432).

“In addition, 5-Aza-dC treatment was generally inefficient in removing aberrant methylation levels in the gene-edited clones, hence suggesting that it may be more complicated than originally thought to remove abnormal modifications from patients’ cells even when combining gene editing with chemical-based approaches.” (Page 19, Lines 509-512).

8. Line 412: “Analysis of DNA methylation levels in the resulting iPSCs was carried out as described for the gene-edited myoblasts by utilizing a non-CpG informative SNP within the DMR (rs635299, see also (12)).” The SNP (rs635299) was referred to reference #14 in line 358. Which is correct?

Response: We apologize for this typo. We corrected the reference to Yanovsky-Dagan (14).

9. Lines 412-414: “This procedure confirmed that reprogramming the gene-edited myoblasts reduced abnormal methylation levels to 93%-0% (Figures 3b and S4b).” The methylation percentage is highly variable among the repeat-less isogenic iPSC clones. The claims in lines 423-427, “As expected, the findings showed an opposite relationship between the extent of residual methylation and TET1 mRNA levels...”, are not convincing because the authors draw this conclusion based upon only IPS3.

Response: We accept the reviewer’s concern regarding the weak correlation between TET1 mRNA levels and the extent of residual methylation in the edited iPSCs. We therefore removed the TET1 qRT-PCR experiment from the manuscript (Fig 3b in the previous version) and only refer to its potential involvement in the undifferentiated state when discussing the proposed model (Fig 4).

10. Lines 477-479: “We found an inverse relationship between the extent of residual hypermethylation and the levels of TET1 mRNA in the gene-manipulated iPSCs.” It is difficult to conclude this from Figure 3b.

Response: See our response to the previous section.

11. Lines 500-509: “We hypothesize that the 3’-end of DMPK, which resides within a CpG island and next to ...” This part of the manuscript is purely speculative. The authors must directly test their hypothesis in their cell models.

Response: Since we removed the qRT-PCR experiment for TET1 from the manuscript, this comment is no longer relevant.

12. Figure 3b: what do the error bars indicate?

Response: Since we removed the qRT-PCR experiment for TET1 from the manuscript, this comment is no longer relevant.

13. Figure S2b: some of the sequences of CL29 are not aligned accordingly.

Response: We have now properly aligned the sequences of predicted off- targets for CL29, as you were right to point out.

14. The authors should clearly mention that they used immortalized myoblasts in the Materials and Methods rather than Acknowledgement.

Response: We now clearly mention that the myoblast cell lines were immortalized by overexpression of hTERT and CDK4 and provide the appropriate reference for this in the EXPERIMENTAL PROCEDURES under the section “Cell Culture”. (Page 5, Line 112). We also refer to this in the Discussion when addressing the issue of residual methylation in the corrected iPSCs:

“Although we predicted that methylation would be completely abolished in the repeat-less iPSCs, it was more frequently reduced rather than fully eliminated. One potential explanation for the apparent variability in residual methylation among the corrected iPSCs may be random integration of the lenti-viral OSKM vector, which offers higher reprogramming efficiency but may result in wide epigenetic variability across clones (33). A different explanation may be the origin of the myoblast cell lines which were established by immortalization with hTERT and CDK4 and may be refractory to reprogramming (34). On the other hand, it has been shown that some loci have a strong propensity to be insufficiently or aberrantly reprogrammed, thus providing hotspots for aberrant epigenomic reprogramming (35).” (Page 20, 521-530).

Reviewer #2 (Remarks to the Author):

Overall, this is a well performed study with a large amount of data that will be of interest for readers in chromatin biology and trinucleotide repeat diseases. However, there are several aspects that would need to be further addressed before conclusions can be drawn.

Specific points:

1. The introduction informs that repeat expansion leads to a autosomal dominant disease allele. It is not clear how DNA methylation and potentially silencing of the downstream gene are related to the disease phenotype. Is heterochromatinization causing a haploinsufficiency or acting in synergy with a pathogenic triplet expanded RNAs? Importantly, would a disease phenotype be expected with the expanded repeat is removed irrespective if heterochromatin remains? This could have implications in the relevance of the heterochromatinization aspect for treatment. If persisting heterochromatin were not causing a phenotype after repeat deletion potentially engineered myoblasts could be considered for treatment. It would be good if this could be clearly state in the introduction or discussion to inform the reader.

Response: Regarding *DMPK* and *SIX5* expression levels, we now show by RT-ddPCR that the wild type and affected DM1 hESCs do not differ in terms of the total *DMPK* and *SIX5* mRNA levels between wild type and DM1 affected hESCs (Figure S3d). Nor do they change as a result of CTG excision when comparing DM1 unmanipulated vs. gene-edited hESCs (Figure S3d). This, in conjunction with our published data regarding no change in mRNA levels between DM1 unmanipulated and gene-edited myoblasts (see Van Agtmaal et al. 2017), strongly suggest that local gene expression is unaffected by abnormal methylation at the DM1 locus, at least not in undifferentiated embryonic cells, as previously claimed.

We now state this in the text:

"It has been claimed that the change in chromatin structure may alter local gene transcription at the DM1 locus (14, 31, 32). Therefore, we assessed the total mRNA levels of *DMPK* and *SIX5* in the wild type, DM1-affected and DM1- Δ/Δ hESCs by RT-ddPCR. The results showed that the wild type and affected DM1 hESCs did not significantly differ for total *DMPK* and *SIX5* mRNA levels (Figure S3d). Nor did they change as a result of CTG excision when comparing DM1 unmanipulated vs. gene-edited hESCs (Figure S3d). Thus, in conjunction with our published data reporting no change in *DMPK* and *SIX5* mRNA expression levels in DM1 unmanipulated as compared to gene-edited myoblasts (see (27)), this strongly suggests that local gene expression is unaffected by abnormal methylation at the DM1 locus, at least not in embryonic stem cells and patients' myoblasts." (Page 15, Lines 377-387).

2. The authors use previously established myoblast lines where an expanded repeat has been deleted and persistence of heterochromatin has been observed. Importantly, they explore if heterochromatin can be reversed by iPSC reprogramming. Although this experiment has a strong expectation that in a pluripotent cell elimination of the repeat should allow reprogramming, the outcome in figure 3 and text line 414 "level to 93%-0%" provide a mixed view. At first glance the 93%-0% appears as a type but at closer inspection it appears that the authors are serious about these numbers. Since 93% and 0% are nearly and exactly at the end of the possible scale of outcomes one would have expected a different discussion. The data clearly show that whereas clone M4 can reverse heterochromatin, clone M6 does not. An explanation such as M6 was not fully reprogrammed is needed to resolve this issue. In my mind

either the authors hypothesis that pluripotent cells always reverse DNA methylation after repeat excision or the myoblasts used are problematic.

Response: Although we predicted that methylation would be completely eliminated by reprogramming into iPSCs, in fact the methylation levels often simply dropped, reaching 0% in only one iPSC clone. The apparent variability in residual methylation among the corrected iPSCs suggests that reprogramming is not uniformly efficient in resetting the epigenetic status of the locus. The reason for this may be one of several: the random integration of the lenti-viral based reprogramming vector, which offers higher reprogramming efficiency but may result in wide epigenetic variability across clones, the immortalized origin of the myoblast cell lines, which may be refractory to reprogramming, and the tendency of specific loci to be insufficiently reprogrammed, providing hotspots of aberrant methylation in human iPSCs.

We now state this in the Discussion:

“Although we predicted that methylation would be completely abolished in the repeat-less iPSCs, it was more frequently reduced rather than fully eliminated. One potential explanation for the apparent variability in residual methylation among the corrected iPSCs may be random integration of the lenti-viral OSKM vector, which offers higher reprogramming efficiency but may result in wide epigenetic variability across clones (33). A different explanation may be the origin of the myoblast cell lines which were established by immortalization with hTERT and CDK4 and may be refractory to reprogramming (34). On the other hand, it has been shown that some loci have a strong propensity to be insufficiently or aberrantly reprogrammed, thus providing hotspots for aberrant epigenomic reprogramming (35). Nevertheless, the general tendency for methylation to be lost by reprogramming of gene-corrected, as opposed to uncorrected, iPSCs further strengthens the hypothesized role of differentiation as a blocking mechanism for the reversibility of this process.” (Page 20, Lines 521-533).

3. The authors argue for a shift from reversible DNA methylation in early embryonic (pluripotent) cells to non-reversible DNA methylation in more differentiated cells (MEFs, myoblasts) based on their observations after CRISPS mediated repeat excision. This is very reasonable and in line with findings in other systems. However, there could also be an influence of the culture conditions. It is known that cells lines in traditional cell culture media a prone to get non-physiological methylation and this could contribute at DM1. To investigate if other regions in the genome might have become aberrantly methylated due to culture conditions independent of the DM1 trinucleotide expansion, control CpG island methylation need to be determined of genes that are normally not methylated but have been shown to become methylated in cultured cell lines. In addition, a comment on the presence of media components that are known to enhance demethylation in the different (iPSC, MEF, and myoblast) media should be considered in the discussion.

Response: It is very unlikely that hypermethylation at the DM1 locus in the affected myoblasts represents a “non-physiological” event caused by culture conditions. This is because this region never methylates in myoblasts unless it harbors a large CTG expansion (based on the cumulative analysis of 150 different wild type molecules). In fact, our analysis

of an informative SNP in the DMR clearly shows that hypermethylation at the DM1 locus is exclusive to the mutant allele, before and after editing.

We believe that the best control CpG island to monitor for “non-physiological methylation” at the mutant locus would be the one that is located precisely at the same position in the genome against the background of a wild type allele. We found that hypermethylation is not acquired in any wild type hESCs or against the background of variant G in affected myoblasts (representing the normal allele). In other words, our bisulfite allele-specific sequencing in the mutant myoblasts (unmanipulated and gene edited) constitutes strong evidence refuting this supposition. Although we do not present the results of the bisulfite sequencing against the background of the wild type allele in the myoblasts (variant G consistently unmethylated), these can be provided upon request. We now highlight this point in the text by stating:

“In no case were normal alleles hypermethylated in the wild type control myoblasts or in the mutant myoblasts against the background of the normal allele (variant G, based on the analysis of 150 wild type molecules in total, data not shown), thus ruling out the possibility of non-physiological hypermethylation due to culture conditions.” (Page 16; Lines 409-413).

In addition, the legend to Figure 2a states that: “None of the normal alleles (variant G) were methylated (data not shown).” (Page 23, Line 624).

To summarize, we have made a major revision to the manuscript, and have addressed all the major and minor concerns raised by the referees. We hope that this manuscript now meets the standards of *Nature Communications*.

Sincerely,

Eiges Rachel, PhD
Head, Stem Cell Research Laboratory
Medical Genetics Institute
Shaare Zedek Medical Center, Jerusalem, Israel.

REVIEWER COMMENTS

Reviewer #1 (Remarks to the Author):

While the authors have made efforts to respond to the previous reviewers' critiques, the current manuscript contains many serious concerns. The authors analyzed methylation of the CTCF binding region in only hESCs, but the description of CTCF binding is inaccurate. The authors conclude that "Altered Chromatin Structure Restores CTCF Binding (line 362 in the revised manuscript)." However, it is difficult to conclude this from Figure S3, where appropriate multiple comparison procedures have not been made. Indeed, the authors wrote that "Although hypermethylation in region F is exclusive to expanded alleles (Figure S3b), CTCF enrichment levels are statistically indistinguishable across all cell types when compared to DM1 hESCs (DM1) (Figure S3c)" in the rebuttal. The authors have opted not to extend studies on methylation and TET1 mRNA expression but rather omitted data in the revised manuscript. It is not a good practice to remove all data that was questioned unless there are some solid reasons. Overall, only methylation and a few chromatin modifications in artificial cellular systems, such as immortalized myoblasts, are reported in this revised manuscript, and the mechanism and downstream effects of methylation or heterochromatinization in DM1 have not been studied. Without further mechanistic insight or some phenotypic analysis, the advance is insufficient for the broad readership of Nature Communications.

Reviewer #2 (Remarks to the Author):

The revised version of the manuscript entitled "Differentiation Shifts from a Reversible to an Irreversible Heterochromatin State at the DM1 locus" by Handal et al. contains additional experiments and clarifications to the text that have improved the study. In particular, the authors now explain the variability of heterochromatin removal in edited DM1 myoblasts by reprogramming in the discussion. The role of culture induced methylation is also addressed in a satisfactory manner by the observation that the unexpanded alleles are never methylated. Although, a sequence that is known to be subject to culture induced methylation might have been more sensitive to detect effects of culture media or conditions that potentially might act on top or in parallel with the physiological DM1 methylation mechanism. During the revision the authors have further identified H3K27me3 as an additional chromatin modification that is implicated in heterochromatin formation at DM1. Taken together the revised manuscript contains a further improved study that has addressed all of my earlier concerns in a satisfactory manner. The study will be of interest to researchers in chromatin biology and trinucleotide expansion disease mechanisms.

We thank Reviewer 1 and Reviewer 2 for their time and effort.

We are extremely grateful to Reviewer 1 for the constructive guidance concerning data analysis and interpretation. Reviewer's 1 insistence that we should expand our mechanistic insights pertaining to hypermethylation dynamics at the DM1 locus in undifferentiated cells aligns perfectly with our ongoing research objectives. As a result, we have undertaken a substantial revision of the manuscript to incorporate these crucial improvements.

We thank Reviewer 2 for acknowledging the progress made in previous revisions and underscoring the significance of our manuscript within the realm of chromatin biology and trinucleotide expansion disease mechanisms. Reviewer 2 did not ask for further revisions.

Hence, below we present a comprehensive point-by-point response to Reviewer 1's feedback, which has undeniably strengthened the overall quality of our work.

Reviewer 1 (Remarks to the Author):

1. The authors analyzed methylation of the CTCF binding region in only hESCs, but the description of CTCF binding is inaccurate. The authors conclude that "Altered Chromatin Structure Restores CTCF Binding (line 362 in the revised manuscript)." However, it is difficult to conclude this from Figure S3, where appropriate multiple comparison procedures have not been made. Indeed, the authors wrote that "Although hypermethylation in region F is exclusive to expanded alleles (Figure S3b), CTCF enrichment levels are statistically indistinguishable across all cell types when compared to DM1 hESCs (DM1) (Figure S3c)" in the rebuttal.

Response: We greatly appreciate the reviewer's valuable input regarding the challenge to conclusively demonstrate the restoration of CTCF binding in the repeat-less DM1 hESC clones through conventional CTCF ChIP experiments. We fully acknowledge the limitation of this approach in the text and have taken proactive measures to effectively address this concern.

In response to the reviewer's insightful comment, we have conducted CTCF ChIP-seq experiments to re-evaluate CTCF binding near the CTG repeats in both DM1 and normal hESCs (at least 3 biological replicates each). Unfortunately, this analysis did not yield any statistically significant differences in CTCF binding between the two cell lines, which aligns with our initial findings (the ChIP-seq data can be made available upon request).

Independently, we leveraged the 2bp deletion that was induced by gene editing at the junction of one of the alleles in CL29 (a DM1 Δ/Δ hESC clone) to show the presence of both alleles in both the input and CTCF-bound fractions following ChIP analysis (Figure S3d). This, together with the known role of methylation in abolishing CTCF binding close the repeat in hESCs (Fig S5D and S5E in Yanovsky-Dagan et al. 2015), and the compelling evidence that gene editing alone does not disrupt CTCF binding next to the CTG (shown by substantial CTCF enrichments in CL9 and CL29 Δ/Δ clones, see Figure S3c), strongly suggests that the loss of heterochromatin through gene editing restores CTCF occupancy on the mutant allele.

In response to the reviewer's valid concern and in alignment with our commitment to accurately reflect our findings, we have proactively adjusted the tone by revising both the chapter title and the accompanying text. The revised text with title now reads as follows:

"Altered chromatin structure likely restores CTCF binding but does not affect local gene transcription.

It has been suggested that aberrant methylation patterns upstream of the CTGs might alter the chromatin structure through the loss of CTCF binding. Consistent with this claim, we determined methylation levels by bisulfite colony sequencing immediately upstream to the repeat (region F in (14)) and validated the binding of CTCF in an overlapping region (CTCF binding site I, CTCFI) by ChIP analysis in wild type, DM1 and Δ/Δ hESCs (Fig S3b and S3c). Although we were unable to distinguish between the wild type and the affected hESCs in terms of the extent of enrichment using this ChIP assay (maximum 2-fold change), we found that hypermethylation at the CTCF binding site was exclusive to DM1 hESCs (Fig S3b). In addition, we leveraged the 2 bp deletion that was induced by gene editing at the junction of one of the alleles in CL29 (unmethylated Δ/Δ clone) to show the presence of two alleles in the CTCF bound fraction after ChIP experiment (Figure S3d). This, together with the known role of methylation in abolishing CTCF binding next to the repeat in hESCs (Fig S5D and S5E in (14)), and the clear evidence for that gene editing does not disrupt CTCF binding next to the CTG (as illustrated by substantial CTCF enrichments in Δ/Δ clones, Figure S3c), strongly suggests that the loss of heterochromatin by repeat deletion restores CTCF occupancy on the mutant allele." (Page 14, Lines 373-390).

"Figure S3 - Validating complete hypomethylation, CTCF binding and local gene transcription in DM1 hESCs before and after gene editing.

d DNA Sanger sequencing of PCR products covering the breakpoint, following CTCF ChIP experiment in CL29 cells demonstrates the presence of two different alleles in both the input and bound fractions. Sanger sequencing of each allele independently, after cloning of PCR products into a TA-vector, served as a control".(Page 38, Lines 1057-1060).

Discussion: "...We show that abnormal methylation and H3K9me3 enrichments are completely abolished by repeat excision in undifferentiated DM1-affected hESCs,

providing evidence for a switch from a closed (heterochromatin) to an open (euchromatin) chromatin configuration which is, at least in part, consistent with similar experiments on iPSCs from fragile X syndrome (5, 6). This is most likely coupled with the rescue of CTCF binding near the repeat (Figure S3c and S3d). Nevertheless," (Pages 21, Lines 584-589).

We firmly believe that this revision not only appropriately addresses the reviewer's comment but also provides more clarity regarding our approach to evaluating CTCF binding within the context of our study.

2. The authors have opted not to extend studies on methylation and TET1 mRNA expression but rather omitted data in the revised manuscript. It is not a good practice to remove all data that was questioned unless there are some solid reasons.

Response: We appreciate the reviewer's concern regarding the exclusion of data related to the levels of *TET1* mRNA expression from the revised manuscript. We understand the importance of transparency and thorough reporting in scientific research and would like to provide a more detailed explanation for our decision.

Upon the initial feedback from the reviewers, we conducted a thorough reanalysis of *TET1* expression within the context of our gene-edited iPSC clones. However, during this re-evaluation, it became evident that *TET1* expression levels exhibited significant variability and inconsistency when compared to the levels of residual methylation observed post-reprogramming. This incongruence raised concerns about the reliability of these findings.

In light of this inconsistency and in an effort to maintain the highest scientific rigor, we made the decision to omit this particular aspect from the manuscript. Our intention was not to withhold data without a valid reason but rather to ensure that the reported results were robust and accurately reflected the outcomes of our study.

We genuinely appreciate the reviewer's commitment to upholding good scientific practice, and we believe that the revised manuscript better aligns with the standards of scientific integrity.

3. Overall, only methylation and a few chromatin modifications in artificial cellular systems, such as immortalized myoblasts, are reported in this revised manuscript, and the mechanism and downstream effects of methylation or heterochromatinization in DM1 have not been studied. Without further mechanistic insight or some phenotypic analysis, the advance is insufficient for the broad readership of Nature Communications.

Response: We appreciate the reviewer's insightful comment on the need for a deeper understanding of the mechanistic features or downstream effects of methylation and heterochromatinization in the context of DM1. To address this concern and provide a more comprehensive analysis, we have extended our research efforts as follows:

1. Identification of differentially expressed chromatin modifiers between undifferentiated hESCs and differentiated cells: We conducted an extensive investigation to uncover the underlying distinctions between undifferentiated hESCs and terminally differentiated cells (TOFs and myoblasts) in terms of aberrant methylation maintenance. To do so, we performed RNA-seq experiments to assess the RNA expression profile of nearly 150 chromatin modifier genes in the genome (see Volcano plots, Figure 4a and Table S4). (Page 18, Lines 495-501).
2. Key Findings: Remarkably, our analysis revealed substantial differences between these two cellular states. Notably, two key groups of enzymes, namely, the *de novo* DNA methyltransferases *DNMT3a* and *DNMT3b*, along with the demethylating enzymes *TET1* and *TET2*, emerged among the most significant differentially expressed genes (DEGs). *DNMT3b* and *TET1* exhibited particularly high and exclusive expression levels to undifferentiated hESCs, whereas *DNMT3a* and *TET2* showed remarkable increased or decreased expression, respectively. (Pages 18-19, Lines 502-514).
3. Functional Validation: To elucidate the functional role of these enzymes, we initiated knockout experiments. Initially, we targeted *DNMT3b*, since it emerged as the most statistically significant and highly expressed chromatin modifier in undifferentiated hESCs. Surprisingly, the *DNMT3b*-null DM1 hESC clones did not exhibit discernible changes in aberrant methylation levels when compared to unmanipulated matched controls (Figure 4c). Given these results, we proceeded to knock out *DNMT3a* in the context of pre-existing *DNMT3b*-null DM1 hESCs to investigate the possibility of functional redundancy between *DNMT3a* and *DNMT3b*, bearing in mind that almost all DMRs in the genome (96%) are redundantly targeted by both enzymes and only lose methylation when both are ablated in hESCs (Liao et al. Nat Genet. 2015). Encouragingly, this approach resulted in a significant decline in abnormal methylation levels in 4 out of the 5 assayed undifferentiated hESC clones, with reductions reaching 18% to 38.5% of the maximum 50% (Figure 4e). (Pages 19-21, Lines 515-569).
4. Conclusions: The findings from this set of experiments underscore the crucial role of *de novo* DNA methyltransferases, *DNMT3a* alone or more likely in tandem with *DNMT3b*, in sustaining abnormal methylation patterns in undifferentiated DM1 hESCs. Importantly, they reaffirm a fundamental difference in the way methylation is preserved in undifferentiated cells as compared to differentiated cells, where *DNMT1* activity plays a more predominant role. This is because abnormal methylation in TOFs and myoblasts remains unaffected following repeat deletion, despite the silencing of *DNMT3b* and/or significant reduction in *DNMT3a* levels by differentiation (as evidenced by RNA-seq, Figure 4a). It is worth noting that these results align with previous research indicating that these enzymes can serve a dual function of initiating and preserving methylation patterns under specific conditions (Walton et al. Epigenetics. 2011 and Liang et al. Mol Cell Biol. 2002).

Altogether, we believe that these additional experiments and findings significantly enhance the mechanistic insights into methylation dynamics in DM1, thus addressing the reviewer's valid concern and contributing to a more comprehensive understanding of the topic.

A detailed list of modifications we made to the manuscript:

1. Abstract: We changed the abstract so that it now includes the new findings related to the role of the *de novo* DNMTs in uniquely maintaining abnormal methylation at the DM1 locus in undifferentiated cells.
2. Materials and Methods: We removed the part that relates to the selection of gRNAs and their validation in 293T cells. In addition, changed the order of sections (cell culture, reprogramming of myoblasts and teratoma production) and added new sections for the RT-qPCR, RNA-seq and Western blot analysis experiments.
3. Results: We now present the new data related to the role of the *de novo* DNMTs in undifferentiated DM1 hESCs as Figure 4 and S7. The proposed model is now Figure 5.
 - a. Deletion of a large CTG repeat expansion from *DMPK* in DM1 hESCs: Removed the introductory part that relates to how the gRNAs were selected and validated in 293T cells.
 - b. "Altered chromatin structure likely restores CTCF binding but does not affect local gene transcription": Although we were unable to distinguish between the wild type and affected hESCs in terms of extent of enrichment using the ChIP assay (maximum 2-fold change), we found that hypermethylation at the CTCF binding site was exclusive to DM1 hESCs (Figure S3b). In addition, we leveraged the 2 bp deletion that was induced by gene editing at the junction of one of the alleles in CL29 (unmethylated Δ/Δ clone) to show the presence of two alleles in the CTCF bound fraction after ChIP experiment (Figure S3d). This, together with the known role of methylation in abolishing CTCF binding next to the repeat in hESCs (Figure S5D and S5E in Yanovsky-Dagan et al. 2015 (14)), and the clear evidence that gene editing does not disrupt CTCF binding next to the CTG (as illustrated by substantial enrichments in Δ/Δ clones, Figure S3c), strongly suggests that the loss of heterochromatin by repeat deletion restores CTCF occupancy in the mutant allele." (Page 14, Lines 373-390; Page 21, Lines 588-589; Page 38, Lines 1057-1060).
 - c. Added new data regarding the fundamental difference in the dynamics of abnormal methylation at the DM1 locus in mutant hESCs vs. differentiated cells (now Figure 4 and S7):
"Abnormal methylation at the DM1 locus is maintained by *de novo* DNMT activity in hESCs:

Given the abovementioned results, we aimed to identify chromatin modifiers that would explain the difference in the maintenance of abnormal methylation at the DM1 locus between undifferentiated and differentiated cells.

First, we profiled gene expression by RNA-seq and compared the undifferentiated DM1 hESCs to their differentiated counterparts, teratoma-derived fibroblasts (TOFs). Furthermore, we extended the analysis to include a comparison of undifferentiated hESCs to the patients' myoblasts, to further substantiate our findings.

After validating the exclusive expression levels of the pluripotent-specific markers *POU5F1* (*OCT4*), *NANOG* and *SOX2* in the hESCs, we compared the expression of nearly 150 potentially relevant genes, collectively representing the complete repertoire of chromatin modifiers in the genome (adapted from dbEM with few additions, see link at <http://web.iiitd.edu.in/rghava/dbem/> and (33)) (Table S4). To visualize the significant differences in the expression of potential chromatin modifiers between the two cell states (hESCs vs. TOFs/myoblasts), Volcano plots were generated (Figure 4a). Based on this analysis, the *de novo* DNA methyltransferases (*DNMT3a* and *DNMT3b*) and the de-methylating enzymes *TET1* and *TET2* emerged as the most significant. While *DNMT3b* and *TET1* exhibited exclusive expression levels in the undifferentiated hESCs, *DNMT3a* and *TET2* demonstrated a marked change, with either increased or reduced expression in the hESCs, respectively. This contrasted with the expression patterns of *DNMT1* and *TET3*, both of which maintained comparable mRNA levels across all cell types and states.

The notable similarity in the differential expression of these enzymes between the two Volcano plots encouraged us to conduct functional assays. First, we chose to knock out *DNMT3b*, because *DNMT3b* stood out as the most statistically significant, up-regulated and exclusively expressed chromatin modifier gene of all the hESCs on the list of differentially expressed genes (DEGs) identified (Figure 4a). By inducing a pair of DSBs with two gRNAs using the CRISPR/Cas9 system, we introduced a homozygous 179 bp deletion overlapping the intron 1-exon 2 boundary in the *DNMT3b* gene (as depicted in Figure 4b). We chose to specifically target exon 2, because it is shared by many different mRNA isoforms of the gene (at least 8 different isoforms for *DNMT3B*). Furthermore, by targeting the 5'-end of the coding region, we increased the probability of introducing premature termination codons (PTCs) or triggering mRNA degradation by nonsense-mediated decay (NMD).

After screening for bi-allelic deletions by PCR (Figure S7a), the potential knockout clones (KOs) were Sanger sequenced (Figure S7b) and then validated by Western blot analysis (Figure 4b). These assays confirmed the complete elimination of the DNMT3b enzyme in 7 out of the 30 transiently selected hESC clones on the background of the DM1 hypermethylated allele.

To evaluate the effect of *DNMT3b* knockout (KO) on aberrant methylation in these cells, we measured the methylation levels at the disease-associated DMR in three randomly selected clones using bisulfite locus-specific DNA deep-sequencing (at least nine cell passages following gene manipulation). However, the *DNMT3b*-null clones did not exhibit any discriminable change in aberrant methylation levels when compared to the unmanipulated matched control (Figure 4c). Given these results, we addressed the possibility of a functional overlap between DNMT3b and DNMT3a enzymatic activity in undifferentiated hESCs, which has been observed in multiple

genomic regions (34). For this purpose, we targeted *DNMT3a* on the background of pre-existing *DNMT3b*-null DM1 hESCs. Using the CRISPR/Cas9 editing system with a pair of gRNAs, we introduced a 128 bp deletion that overlapped with the boundary between exon 19 and the following intron in the *DNMT3a* gene (Figure 4d). This approach was chosen for two main reasons. The first is that *DNMT3a* exhibits multiple isoforms (at least 6) arising from alternative transcription initiation and splicing sites, most of which involve the 3'-end of the gene (exons 7 to 23 region). The second is that exon 19 is critical for the catalytic activity of the enzyme (exons 16 to 20), in that it facilitates the comprehensive elimination of potentially active protein species (35). Based on these rationales, we introduced a bi-allelic deletion in *DNMT3a* on the genetic background of DM1 *DNMT3b* KO hESCs in 5 out of 60 transiently selected puro-resistant clones (Figure S7c).

After screening for homozygote deletions by PCR and Sanger sequencing (Figure S7d and S7e), we performed quantitative reverse transcription PCR (RT-qPCR) to search for a substantial reduction in *DNMT3a* mRNA levels (Figure 4d). We found that gene manipulation indeed led to either the complete loss (CI-I3) or, more commonly, a significant reduction in *DNMT3a* mRNA levels (Figure 4d). However, confirming the absence of DNMT3a enzymatic activity in these cells was difficult due to the absence of a straightforward assay for DNMT3a-specific activity. Nonetheless, given the pivotal role of exon 19 in the catalytic domain of DNMT3a (35), the residual transcripts were likely to be non-functional because of exon 19 skipping, as validated by Sanger sequencing of the RT-PCR products (Figure 4d and Figure S7e). In line with this view, we generated five different bi-allelic *DNMT3a*-double targeted DM1 hESCs.

To assess the effect of *de novo* DNMT3 double targeting, we monitored the levels of abnormal methylation within the disease-associated DMR by utilizing bisulfite locus-specific DNA deep-sequencing, as described above for single *DNMT3b* KO clones. There was a significant reduction in abnormal methylation levels in four out of the five assayed clones, ranging from 18% to 38.5% (out of a maximum of 50%, Figure 4e). Crucially, note that unlike the single *DNMT3b* KOs, most of the double targeted clones became morphologically abnormal, tended to spontaneously differentiate, and proved to be incapable of sustaining growth beyond five passages.

In summary, this study revealed a fundamental difference between undifferentiated and differentiated cells in terms of the role played by *de novo* DNMTs (DNMT3a singly or jointly with DNMT3b) in maintaining abnormal methylation patterns at the DM1 locus in undifferentiated hESCs. Furthermore, and unlike in differentiated cells, this molecular event is dependent on the DNA sequence; i.e., the disease-causing CTG expansion at the *DMPK* gene." (Pages 18-21, Lines 494-569).

4. Discussion: We now review the new data related to the role of the *de novo* DNMTs in uniquely maintaining aberrant methylation patterns at the DM1 locus in undifferentiated DM1 hESCs (vs. differentiated) and discuss its implications. (Pages 23-24, Lines 622-670)

In addition, we deleted the part referring to the resemblance with *Xist*- mediated epigenetic silencing before and after differentiation.

5. Figures and Tables:

- a. New Figure 4: RNA-seq comparisons represented by Volcano plots (4a), *DNMT3b* editing and validation (4b), analysis of DNA methylation by bisulfite locus-specific deep-sequencing in *DNMT3b* KO clones (4c), *DNMT3a* editing in *DNMT3b*-null DM1 hESC clones (4d), analysis of DNA methylation by bisulfite locus-specific deep-sequencing in *DNMT3* double-targeted DM1 hESCs clones (4e).
- b. Figure 5: Proposed model to explain the fundamental difference in the dynamic of DNA methylation between undifferentiated vs. differentiated DM1 cells.
- c. Figure S3d: DNA Sanger sequencing of CTCF input and bound fractions in CL29 (Δ/Δ DM1 hESC clone).
- d. New Figure S7: PCR screen for the identification of CRISPR/Cas9 induced homozygous deletions in *DNMT3b* (S7a), DNA Sanger sequencing for validation of *DNMT3b* bi-allelic targeting (S7b), PCR screen for the identification of CRISPR/Cas9 induced homozygous deletions in *DNMT3a* (S7c), DNA Sanger sequencing for validation of bi-allelic *DNMT3a* targeting (S7d), validation of exon 19 skipping in *DNMT3a* residual RNA by RT-PCR after gene editing (S7e).
- e. Table 2: addition of gRNA and primer sequences.
- f. Table 3: removed hESC line annotations from the original table.
- g. New Table 4: Volcano plots data from the RNA-seq experiments comparing the expression of nearly 150 chromatin modifiers between the different cell types.

The revised version of the manuscript in TRACK CHANGES of all the modifications is attached.

To summarize, we have made a major revision to the manuscript, and have addressed all the major and minor concerns raised by the referees. We hope that this manuscript now meets the standards of *Nature Communications*.

Sincerely,

Eiges Rachel, PhD
Head, Stem Cell Research Laboratory
Medical Genetics Institute
Shaare Zedek Medical Center, Jerusalem, Israel.

REVIEWERS' COMMENTS

Reviewer #1 (Remarks to the Author):

The authors have effectively responded to the concerns raised in the previous round of critiques. While certain aspects persist in a confirmatory nature owing to experimental constraints, the study should prove to be of considerable interest to the field.